# CALC-2020: a new baseline land cover map at 10 m resolution for the circumpolar Arctic

Chong Liu[1], Xiaoqing Xu[2], Xuejie Feng[1], Xiao Cheng[1], Caixia Liu[3], Huabing Huang[1,2,4]

[1]School of Geospatial Engineering and Science, Sun Yat-Sen University, and Southern Marine Science and Engineering Guangdong Laboratory (Zhuhai), Zhuhai 519082, China
[2]Peng Cheng Laboratory, Shenzhen 518066, China
[3]State Key Laboratory of Remote Sensing Science, Aerospace Information Research Institute, Chinese Academy of Sciences, Beijing 100101, China
[4]International Research Center of Big Data for Sustainable Development Goals, Beijing 100094, China

*Correspondence to*: Caixia Liu (liucx@radi.ac.cn) and Huabing Huang (huanghb55@mail.sysu.edu.cn)

**Abstract.** The entire Arctic is rapidly warming, which brings in a multitude of environmental consequences far beyond the northern high-latitude limits. Land cover maps offer biophysical insights into the terrestrial environment and are therefore essential for understanding the transforming Arctic in the context of anthropogenic activity and climate change. Satellite remote sensing has revolutionized our ability to capture land cover information over large areas. However, circumpolar Arctic-scale fine resolution land cover mapping has been so far lacking. Here, we utilize a combination of multimode satellite observations and topographic data at 10 m resolution to provide a new baseline land cover product (CALC-2020) across the entire terrestrial Arctic for circa 2020. Accuracy assessments suggest that the CALC-2020 product exhibits satisfactory performances, with overall accuracies of 79.3% and 67.3%, respectively, at validation sample locations and field/flux tower sites. The derived land cover map displays reasonable agreement with pre-existing products, meanwhile depicting more subtle polar biome patterns. Based on the CALC-2020 dataset, we show that nearly half of the Arctic landmass is covered by graminoid tundra or lichen/moss. Spatially, the land cover composition exhibits regional dominance, reflecting the complex suite of both biotic and abiotic processes that jointly determine the Arctic landscape. The CALC-2020 product we developed can be used to improve earth system modelling, and benefit the ongoing efforts on sustainable Arctic land management by public and non-governmental sectors. The CALC-2020 land cover product is freely available on Science Data Bank: http://cstr.cn/31253.11.sciencedb.01869 (Xu et al., 2022a).

## 1 Introduction

Accounting for ~5.5% of the Earth's land surface, the Arctic disproportionately affects global biogeochemical cycles (Jeong et al., 2018; Landrum and Holland, 2020; Miner et al., 2022) and harbours a large proportion of high-latitude biodiversity (Niittynen et al. 2018; Christensen et al. 2020). During the past decades, the Arctic as a whole is rapidly warming (Previdi et al. 2021), with crucial consequences in the terrestrial section including land ice retreat (Shepherd et al., 2020), permafrost thawing (Hjort et al., 2018), vegetation greening/browning (Myers-Smith et al., 2020; Berner et al., 2020; Bartsch et al., 2020a),

and intensified greenhouse gas emissions (Najafi et al., 2015; Descals et al., 2022). These changes have profound impacts on Arctic biomes (Hodkinson et al., 1998; Shevtsova et al., 2020; Wang and Friedl, 2019), and put millions of local residents and their cultures at risk (Huntington et al., 2019). Moreover, a changing Arctic is increasingly influencing human societies outside

of the Arctic (Moon et al., 2019), through sea level rise and atmospheric circulation. Without effective strategies for mitigating Arctic environmental changes, the goal of global sustainable development remains elusive (Beamish et al., 2020; Liu et al., 2021a, 2022).

As a key terrestrial surface descriptor, land cover is central to our understanding of the changing Arctic (Bartsch et al., 2016; Liang et al., 2019; Raynolds et al., 2019; Wang et al., 2020). The land cover regulates the surface energy fluxes, which

contribute to climate change and, in turn, influence land surface properties and the provision of ecosystem services (Friedl et al., 2010; Gong et al., 2013; Wulder et al., 2018; Song et al., 2018). Given the Arctic's ecological importance, some earlier efforts have been made to map Arctic land cover based on field investigations (Ingeman-Nielsen and Vakulenko, 2018; Lu et al., 2018) or existing atlases (Walker et al., 2005; Raynolds et al., 2014), both of which are nevertheless laborious, time consuming and resource demanding. With synoptic view and repeatable coverage, satellite observations provide an

unprecedented way to delineate and analyse Arctic land cover at multiple scales. A few studies attempted to capture Arctic land cover using satellite remote sensing, with observations obtained from satellites of Landsat (Jin et al., 2017; Wang et al., 2020), SPOT (Kumpula et al., 2011), and Sentinel-2 (Bartsch et al., 2020b). But these studies focused mainly on small areas, being unable to provide spatially complete information for the entire terrestrial Arctic. In parallel, some existing scientific programs have manifested remarkable achievements of general-purpose land cover maps at the global scale, including (part

of) the Arctic region (Loveland et al., 2000; Friedl et al., 2010). However, these products bear with coarse spatial resolutions (100 m~1 km pixel size), hence raising the sub-pixel mixing issue (Friedl et al., 2022). The recent advances in satellite data accessibility offers a new possibility to explore large-area environmental change (Gong et al., 2013). Different from traditional products derived from coarse-resolution imagery, now fine-resolution (10~30 m pixel size) land cover datasets become available at continental to global scales.

Although the entire Earth surface witnessed a growing number of fine spatial resolution land cover products (Gong et al., 2013; Chen et al., 2015; Karra et al., 2021; Zanaga et al., 2020; Brown et al., 2022), most of them have systematically low accuracy in Arctic (Bartsch et al., 2016; Liang et al., 2019), thus not fully meet the need for precise Arctic land cover distribution and composition information. The terrestrial Arctic environment is a fundamentally different ecosystem from those at lower latitudes, and this calls for reconsideration of land cover mapping paradigm from alternative aspects including classification

legend design, remote sensing data acquisition and computing performance. For example, the lichen/moss biome is extensively distributed within high-latitude ecozones, but such a type is absent in most land cover classification schemes (Friedl et al., 2022). Moreover, the common presence of treeless tundra landscape patches gives rise to the "spectral confusion" issue that can lead to a decreased classification accuracy in the Arctic (Liang et al., 2019; Bartsch et al., 2020b). Severe cloud contamination and high solar zenith angles also introduce uncertainties into the results derived from optical imagery (Berner

et al., 2020). Hence, efforts of mapping circumpolar Arctic land cover should be complemented by information beyond the

spectral domain. Space-borne Synthetic Aperture Radar (SAR) is capable of penetrating clouds and thus providing valuable earth observation information when and where valid optical image data are insufficient (Engram et al., 2013). Recent studies suggested that the inclusion of SAR data is essential for generating spatially continuous map of land cover within the Arctic (Bartsch et al., 2020b, 2021). In addition to optical and SAR data, terrain coefficients can also facilitate the identification of Arctic biomes by incorporating environmental factors including temperature, solar radiation, and water availability (Raynolds et al., 2019). Furthermore, the rapid development of cloud computing platforms, such as Google Earth Engine (GEE) (Gorelick et al., 2017), Amazon Web Services (AWS) (Liu et al., 2021b) and NASA Earth Exchange (NEE) (Nemani et al., 2010), enables taking full advantage of geo-big data, thus making multi-source based circumpolar Arctic land cover mapping feasible. In this study, we present a new circumpolar Arctic land cover product for circa 2020 (CALC-2020 hereafter), through synergistically integrating multimode remote sensing data captured by the Sentinel satellite sensors and terrain layers derived from recently-released ArcticDEM. Within the Arctic extent, each land pixel is characterized by its dominant biophysical component using a modified FROM-GLC classification scheme, at 10 m spatial resolution. To create the CALC-2020 map, metrics were derived from Sentinel-1 polarisation, Sentinel-2 surface reflectance and ArcticDEM topographic bands, serving as input features for a machine-learning classification procedure based on the GEE platform. The classification model was locally calibrated with a training sample set collected from multiple data sources. We aim, by resolving the most updated spatial patterns and composition of land cover across the terrestrial Arctic, to advance our knowledge of environmental change at northern high latitudes, and to enlighten sustainable land management by public and non-governmental sectors.

## 2 Materials and methods

### 2.1 Study area and land cover classification scheme

There exist various definitions of what extent is contained within the Arctic. In the present study, we delimited the study area in the terrestrial Arctic, following our previous practices by Liu et al. (2021a) and Xu et al. (2022b). Here the terrestrial Arctic is defined as the northernmost part of the Earth characterized by tundra vegetation, an arctic climate and arctic flora, with the tree line and continental coastlines jointly determining the extent borders (**Figure 1**). Spatially, the present study covers an area of approximately 7.11 million km$^2$, overlapping with parts of six countries including Canada (CA), Denmark (Greenland, GR), Iceland (IC), Norway (NO), Russia (RU), and the United States (Alaska, AK). Within the study area, we implemented a ten-category classification scheme to represent the land cover diversity across the terrestrial Arctic. This scheme evolved from the level-1 classification system of FROM-GLC (Finer Resolution Observation and Monitoring of Global Land Cover) version 2017 (Gong et al., 2019), with necessary modifications that adapt to the geographical environment at northern high latitudes (Liang et al., 2019). More specifically, we excluded the grassland class due to its rareness, and subdivided the tundra biome into three categories: graminoid tundra, shrub tundra, and lichen/moss. **Table 1** provides the definition of each land cover type included in the present study.

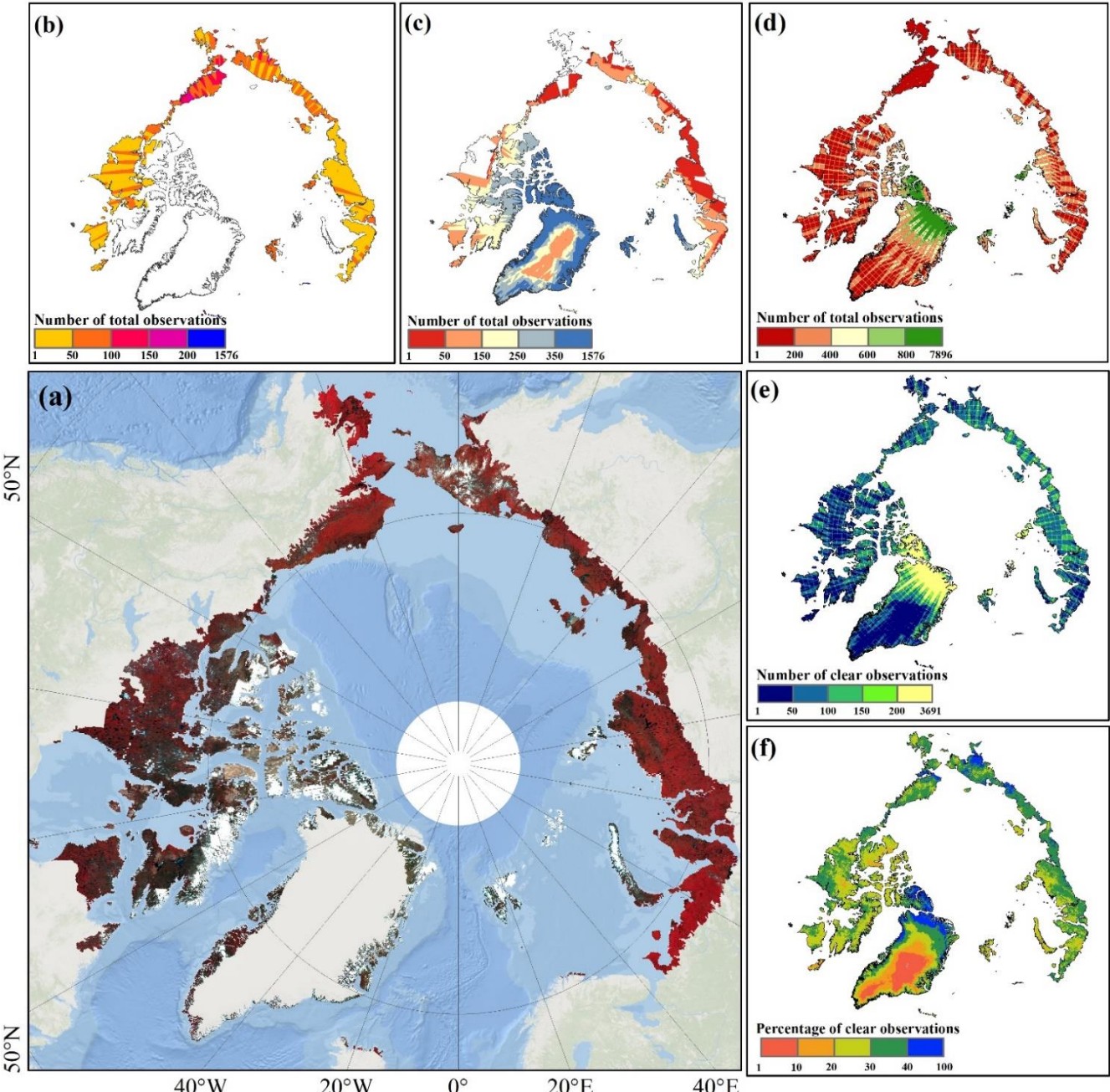

**Figure 1: Overview of the study area.** **(a) Spatial extent of the circumpolar Arctic, located at the northernmost part of the Earth with a total area of ~7.11 million km². Bands 8, 4, 3 are displayed in as red, green and blue layers for the Sentinel-2 composite image. The base map is from ESRI. (b)~(c) show spatial distributions of per-pixel satellite observation availability for Sentinel-1 with VV+VH and HH+HV band combinations, respectively. (d)~(f) are spatial distributions of total observations, clear observations, and clear observation percentage of Sentinel-2.**

**Table 1: Description of the ten-category land cover classification scheme used in the present study.**

| Land cover type (ID) | Description |
|---|---|
| Cropland (CRO) | Arable land that is sowed or planted at least once within a 12-month period, including irrigated or rain-fed field, plantation, and greenhouse |
| Forest (FST) | Land covered by trees, with canopy coverage greater than 30% and canopy height typically no less than 2 m |
| Graminoid tundra (GRT) | Land covered by herbaceous vegetations with plant height typically ranging 5~15 cm |
| Shrub tundra (SRT) | Land covered by shrubs of any stature with plant height typically ranging 20~50 cm |
| Wetland (WET) | Land featured by aquatic plants and periodically saturated with or covered by water |
| Open water (OWT) | Inland open water bodies, including rivers, lakes, reservoirs, pits and ponds |
| Lichen/moss (LAM) | Bedrock covered by cryptogam communities |
| Man-made impervious (MMI) | Impermeable land surface paved by man-made structures |
| Barren (BAR) | Natural dry land with vegetation coverage typically less than 10% |
| Ice/snow (IAS) | Land covered with snow and ice all year round |

## 2.2 CALC-2020 input data

We used the GEE platform to obtain and preprocess remote sensing datasets in this study. All image collections were independently filtered by the extent of the study area and the study period (the year 2020). The Sentinel-1 mission is composed of a constellation of two satellites (S-1A and S-1B), both performing dual-polarization C-band SAR imaging with a 12-day repeat cycle at the equator. Among various Sentinel-1 products, we used the Level 1 Ground Range Detected (GRD) product in the Interferometric Wide (IW) swath mode at 10 m spatial resolution. Given the imbalanced data coverage of polarization combinations across the Arctic (**Figure 1b~c**), we selected dual-band cross-polarization, horizontal transmit/vertical receive bands (HH+HV) for Canada and Greenland, and the dual-band cross-polarization, vertical transmit/horizontal receive bands (VV+VH) for the remaining countries. The Sentinel-2 Multi-Spectral Instrument (MSI) onboard both S-2A and S-2B satellites is an optical sensor having started observing the earth's terrestrial surface since 2015, with a spatial resolution of 10~60 m depending on the wavelength. The present study used the Level 2 surface reflectance product of Sentinel-2 to ensure that geometric and radiometric qualities meet the requirements. For each Sentinel-2 image, three visible bands, four red edge bands, three infrared bands, one scene classification map band (SCL) and one quality assessment band (QA60) were employed. We pan-sharpened the red edge and infrared bands to 10 m using the bicubic interpolation algorithm (Liu et al., 2020) to match the resolution of visible bands. In addition to satellite imagery, we also included the 10 m ArcticDEM digital surface model product in our data pool for characterizing the topographic properties of each Arctic land pixel. Encompassing all land area north of 60°N, the ArcticDEM v3.0 product was generated from very high resolution (VHR) stereo images (Porter et al., 2018).

**2.3 Creation of CALC-2020 map**

Based on input data from Sentinel-1, Sentinel-2 and ArcticDEM, we developed a comprehensive framework (**Figure 2**) to guide circumpolar Arctic land cover mapping and analysis at 10 m spatial resolution for circa 2020. For the purpose of supervised classification model development, a "ready for use" training sample set was constructed derived from multiple sources. We created two separate circumpolar Arctic map products: a map of the man-made impervious surface extent and a map of the natural land cover distribution. For mapping the man-made impervious surface extent, we directly leveraged the existing Circumpolar Arctic Man-made impervious area product (CAMI-2020) from our pilot study (Xu et al., 2022b). For mapping the natural land cover distribution, we developed local adaptive random forest models for each country and performed supervised classification using polarimetric, spectral, phenological and topographic feature metrics. Detailed procedures within the framework are described below.

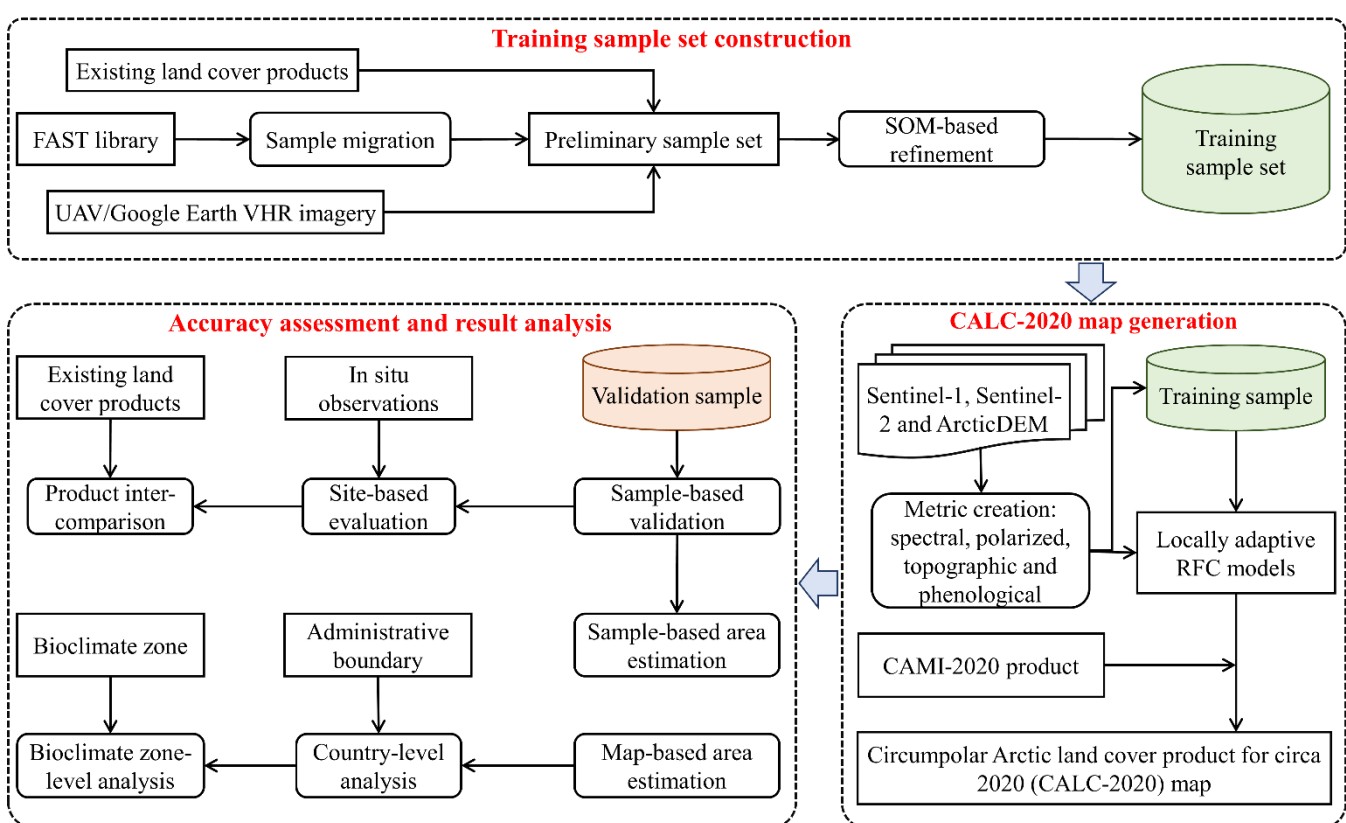

**Figure 2: Framework of creating CALC-2020 map with the use of Sentinel-1, Sentinel-2 and ArcticDEM data.**

**2.3.1 Training sample set construction**

The supervised land cover classification approach requires reliable training sample for model development (Foody et al., 2016; Hermosilla et al., 2022). In the present study, the CALC-2020 training sample set was constructed from three sources (**Figure S1**). First, we used the world's first all-season sample library (FAST) (Li et al., 2017) to generate the backbone of our training

data. FAST offers 91,619 sample locations and their multi-seasonal land cover type information at the planetary scale for circa 2017. We excluded FAST records that are outside of our study area or experienced land cover change(s) by conducting spectral

similarity measurement between the reference year (i.e., 2017) and the target year (i.e., 2020) (Huang et al., 2020). For each retained FAST sample location, we created a $90 \times 90$ m square buffer, in which the land cover labels of all pixels were acquired by leveraging the SCL band layer of Seninel-2 (Liu et al., 2021a). A FAST sample record was preserved only when it represented the dominant land cover type within its buffer area (i.e., greater than 50% area proportion). The above-mentioned procedure resulted in a total of 14,579 preliminary sample records derived from FAST. The second source of training data is

existing land cover maps, including NLCD 2016 (for Alaska), Land Cover of Canada 2015 (for Canadian Arctic), and GlobeLand30 V2020 (for the rest terrestrial Arctic countries). We incorporated these products into one single land cover mapping layer by unifying their classification schemes into the CALC-2020 legend (**Table 1**) based on prior knowledge. For example, wet tundra (GlobeLand30) and grassland/herbaceous (NLCD 2016) are equivalent to wetland and graminoid tundra, respectively, due to their similar definitions. With the re-classified reference land cover layer, sample extraction was performed

by using a stratified random sampling strategy. We randomly collected 12,000 points for each CALC-2020 class, except for cropland (200 points), forest (2,000 points) and shrub tundra (5,000 points) because of their limited area occupations. All extracted points were double checked by senior interpreters to minimize errors associated with the data source. Special care was taken to make a distinction between graminoid tundra and shrub tundra because they are easily confused for a single season. Thus, time series images from Sentinel-1 and Sentinel-2 were used to support judgement as needed. After removing

pixels deemed incorrect, we retained 64,133 preliminary training sample points derived from existing land cover maps. Given the absence of the lichen/moss class in FAST and most existing land cover products, we additionally adopted Google Earth imagery data and UAV aerial images provided by the United States Geological Survey (USGS) as the source of lichen/moss training sample. Spatial/contextual information domains of reference VHR images were used to discriminate lichen/moss from other vegetated covers. The sample size of lichen/moss was 5,000 to balance sampling representativeness and interpretation

workload. We kept only well-interpreted points with high-level confidence, which eventually led to 4,913 preliminary training sample points for lichen/moss.

Due to inherent classification scheme inconsistence and acquisition year mismatch among multiple sources, the preliminary training sample inevitably contains errors that could undermine or even lead to the failure of CALC-2020 mapping. Therefore, we conducted a refinement approach to obtain a "ready for use" training sample set based on the self-organizing map (SOM)

technique. SOM, also known as Kohonen neural network, is an effective and automatic tool for the task of clustering and classification (Kohonen, 2013). It represents the input data distribution by using a two-dimensional map in which models are automatically associated with neurons. In this study, we created and trained a $10 \times 10$ map of 100 neurons using the batch Weight/Bias training algorithm. The map training completes when reaching the maximum number of epochs ($i = 200$). **Figure 3** illustrates the procedure of SOM-based sample refinement. For a given land cover class $C$ (other land cover classes termed

$Rs$), we randomly selected $N$ sample records labelled as $C$ and $2N$ sample records labelled as $Rs$ from the preliminary training sample, respectively. We then grouped the selected sample into 100 clusters, within each of which the purity index was

acquired by calculating the percentage of sample records labelled as $C$. We set a purity index threshold of 75%, aiming to balance sample size and sample robustness (Gong et al., 2019). This approach was applied to all classes for constituting the "ready for use" training sample. After the SOM-based refinement, the final training sample includes 70,260 valid records, including 192, 2,836, 15,686, 4,794, 6,470, 11,729, 4,380, 11,445 and 12,728 points for cropland, forest, graminoid tundra, shrub tundra, wetland, open water, lichen/moss, barren, and ice/snow, respectively.

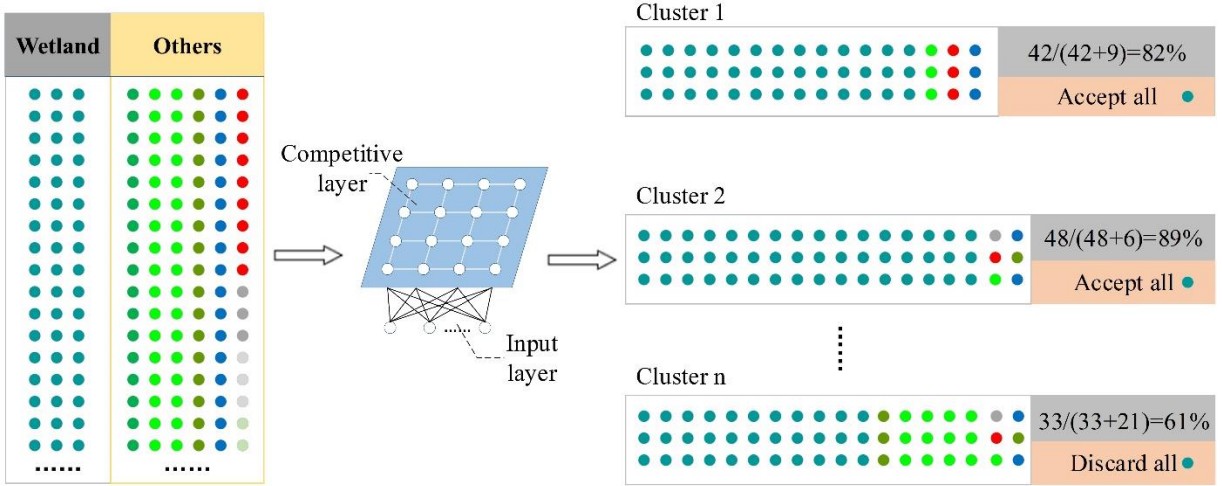

**Figure 3: Illustration of the SOM-based training sample refinement procedure.**

### 2.3.2 Man-made impervious surface mapping

Man-made impervious surface is a representative land cover type indicating human footprint. However, mapping man-made impervious surface has been difficult because it consists of diverse artificial materials and spatial forms (Liu et al., 2022). This issue becomes more prominent in the Arctic where impervious surface clusters are usually fragmented and small in size (e.g. oil/gas deposits). Therefore, we leveraged an existing product, CAMI-2020, from our pilot study (Xu et al., 2022b) to pre-classify Arctic man-made impervious surfaces. CAMI-2020 was developed by integrating satellite imagery and OpenStreetMap as input features, and it provides the first spatially continuous map of Arctic man-made impervious surface distribution at 10 m resolution. Accuracy assessment suggested that the CAMI-2020 map is capable of depicting the spatial pattern of man-made impervious surfaces across the Arctic, with overall accuracy and Kappa value of 86.4% and 0.7, respectively. Due to its robustness, CAMI-2020 was used for mapping the Arctic man-made impervious surface extent in the present study. The CAMI-2020 dataset is publicly available from http://www.doi.org/10.11922/sciencedb.01435.

### 2.3.3 Natural land surface mapping

After acquiring the extent of man-made impervious surface, we conducted natural surface land cover mapping based on feature metrics derived from Sentinel-1, Sentinel-2 and ArcticDEM. For Sentinel-1, a seasonal compositing approach was undertaken to obtain the median value of all observations for growing months (June, July, and August) and dormant months, separately.

For Sentinel-2, we first identified and masked invalid observations, including clouds, cloud shadows, and snow, according to the QA60 band. Then three groups of Sentinel-2 feature metrics were extracted: 1) per-band values representing growing season reflectance using median and greenest compositing methods, respectively; 2) per-index values representing selected percentiles (10%, 50% and 90%); 3) phenometrics including the start of growing season (SOS), end of growing season (EOS), the peak of growing season (POS), and the largest data value of growing season (LDOG). To reduce the impact of noise and data gaps in Sentinel-2 time series, we followed a statistic-based algorithm (Bolton et al., 2020) to estimate the phenometrics (**Figure S2**). Cloud-free Sentinel-2 observations were interpolated in each pixel at an 8-day time step using penalized cubic smoothing splines. With the smoothed, seamless reflectance time series, we calculated the normalized difference vegetation index (NDVI) at each temporal interval to depict the time (day of year, DOY hereafter) of the vegetation phenophase transitions. The maximum value of the smoothed NDVI time series was identified as LDOG. SOS and EOS were retrieved as the DOYs when the NDVI time series cross 50% of the amplitude in the greenup and greendown periods, respectively. POS was identified as the DOY when the NDVI time series reaches LDOG. Topographic metrics were directly computed from ArcticDEM, including elevation, slope, and aspect. In summary, we created a total of 51 feature metrics for natural land surface classification. **Table S1** provides detail descriptions of the metrics used in the present study.

Based on all metric sets described above, we used Random Forest Classifier (RFC) to generate Arctic's natural land cover map. RFC is a non-parametric machine learning method that ensembles a multitude of decision trees for class membership prediction (Breiman, 2001). Compared with other supervised classification algorithms, RFC is more robust in mapping large-area land cover and can accommodate high dimension input features (Zhu et al., 2012; Gong et al., 2020a; Zhang et al., 2021a). For the purpose of balancing classification accuracy and computational efficiency, we parameterized RFC with 500 decision trees and the square root of the total number of input variables as the number of variables to split each node (Liu et al., 2019). RFC model training and prediction were performed individually for each country using the "smileRandomForest" API in GEE.

**2.4 Mapping performance evaluation**

We designed three methods to evaluate the performance of CALC-2020. First, we implemented a stratified random sampling (**Figure S3**) for assessing the accuracy and uncertainty of our estimated land cover map, based on good practices by Olofsson et al. (2014). We used CALC-2020 map itself as the stratification of study area, and set the validation sample size to 6,513 by specifying a target standard error for overall accuracy (OA) of 0.5%. We allocated 40~1872 sample units for each land cover class (see **Section 3.1.1**) and calculated error metrics including user's and producer's accuracies (UA and PA), along with estimates of associated 95% confidence intervals. The reference class label for each sampled pixel was identified based on expert interpretation of cloud-free Sentinel-2 images and Google Earth VHR imagery data, as available. Sample pixels with disagreement among experts were subsequently revisited until a consensus was reached. In the second evaluation method, we examined the CALC-2020 mapping performance using in-situ data obtained from ORNL DAAC's MODIS/VIIRS Land Product Subsets project (ORNL DAAC 2018). We employed the Fixed Sites Subsets Tool to select all field and flux tower sites within our study area (55 sites in total, **Table S2**). For each site, the dominant land cover type was determined by referring

to its meta data as well as the near-surface camera images (if available). The third evaluation method is the comparison of CALC-2020 with three widely-used global fine resolution land cover products: ESA WorldCover V100 (Zanaga et al., 2020), ESRI Global Land Cover (Karra et al., 2021), and GlobeLand30 V2020 (Chen et al., 2015). These products were selected because 1) they have consistent data epochs and adequate spatial resolutions that make them comparable to the CALC-2020 map; 2) they include the majority of land cover types used in CALC-2020 legend, thus the comparing results can be more robust and less affected by the classification scheme discrepancy. It should be noted that either selected land cover products nor CALC-2020 is considered as ground truth. Instead, the inter-comparison provides an overall insight of pixel-level agreement, both statistically and spatially (Liu et al., 2020). At the per-pixel level, the paired land cover comparison result consists of four categories: agreement (AG), disagreement due to model prediction (DM), disagreement due to scheme difference (DS), and disagreement due to data missing (DD). **Table 2** offers the detailed definition of each paired land cover comparison category. As an additional comparison to complement the inter-product evaluation, we used the validation sample shown in **Figure S3** to calculate accuracy metrics of three global land cover products. To harmonize various classification legends to that of CALC-2020, the grass (ESA WorldCover, ESRI Global Land Cover) and wet tundra classes (GlobeLand30) were treated as equivalents of graminoid tundra and wetland, respectively.

Table 2: **Description of per-pixel level comparison categories between CALC-2020 and reference products.**

| Category (abbreviation) | Definition |
| --- | --- |
| Agreement (AG) | CALC-2020 and the compared land cover product display identical classification result |
| Disagreement due to model prediction (DM) | CALC-2020 and the compared land cover product display different classification results, both of which are included in the CALC-2020 map legend |
| Disagreement due to scheme difference (DS) | The compared land cover product displays a classification result which is not included in the CALC-2020 map legend |
| Disagreement due to data missing (DD) | Unclassified or data missing exhibited by the compared land cover product |

**2.5 Land cover area estimation**

We performed land cover area estimation at two stages to ensure the validity of all statistics reported throughout this study. At the first stage, we utilized the error matrix obtained from validation to produce "unbiased" circumpolar Arctic land cover area estimations as well as their uncertainties (95% confidence interval). For each CALC-2020 class, the area estimator is based on the mapping stratum, the proportion estimated from the reference data, and its standard error (Olofsson et al., 2014). Despite the potential of correcting area estimation biases, the sample-based area estimation strategy is highly dependent on sample allocation, which may unnecessarily limit its effectiveness in the Arctic because of the highly imbalanced sample availability among countries and across bioclimate zones (Liu et al., 2022). Therefore, at the second stage, we employed the conventional pixel counting method (Gong et al., 2020a) to calculate area statistics directly from the CALC-2020 dataset. This map-based area estimation strategy is straightforward and flexible at different spatial levels. We treated sample-based and map-based area statistics as complementary metrics for better describing the CALC-2020 derived land cover patterns at multiple levels.

# 3 Results and discussion

## 3.1 Reliability of CALC-2020 map

### 3.1.1 Sample-based evaluation

Following good practices by Olofsson et al. (2014), we built the error matrix and associated sample-based accuracy statistics of the CALC-2020 map based on 6,513 validation points (**Table 3**). The OA of the CALC-2020 dataset for the circumpolar Arctic is 79.3±1.0% (95% confidence interval). At the biome level, we found the classifications of all land cover types have reasonable accuracies, with UA ranging from 74.0±2.7% (ice/snow) to 95.9±5.6% (man-made impervious). Similarly, most biomes exhibit satisfactory PA results (above 75%), except for shrub tundra (47.1±5.7%), open water (62.7±2.7%) and barren (62.2±5.9%) with less desirable results. It should be noted that all metrics reported in **Table 3** are based on error matrix of area proportion, therefore inevitably different with those derived from traditional confusion matrix of sample counts. For example, the traditional confusion matrix-derived PA of shrub tundra is 68.1% (**Table S3**), whereas its stratified error-adjusted PA estimate is lower, due primarily to the influence of estimation weights (area proportions of map classes). Given the generally large reflectance discrepancy between water and non-water covers, the less desirable performance of CALC-2020 in water extraction may seem unexpected. This highlights the distinctiveness of Arctic's geographical environment that can affect the spectral signal of water in space and time (Gong et al., 2016). Specifically, shallow water bodies are easily confused with barren lands because of the mixed pixel issue (**Figure S4a~b**). Moreover, the employed satellite images may only capture the freezing stage for some water pixels, which were misclassified as ice/snow in the CALC-2020 map (**Figure S4c**).

**Table 3: Error matrix of the CALC-2020 map based on validation sample. UA, PA and OA indicate user's accuracy, producer's accuracy and overall accuracy, respectively. Reference classes are in columns. Land cover abbreviations are given in Table 1.**

| Class | CRO | FST | GRT | SRT | WET | OWT | LAM | MMI | BAR | IAS |
|---|---|---|---|---|---|---|---|---|---|---|
| CRO | **43** | 0 | 3 | 0 | 0 | 0 | 0 | 0 | 0 | 0 |
| FST | 0 | **30** | 2 | 8 | 0 | 0 | 0 | 0 | 0 | 0 |
| GRT | 0 | 0 | **1501** | 62 | 81 | 28 | 125 | 0 | 35 | 0 |
| SRT | 0 | 0 | 25 | **160** | 1 | 0 | 2 | 0 | 1 | 0 |
| WET | 0 | 0 | 35 | 5 | **375** | 14 | 30 | 0 | 4 | 0 |
| OWT | 0 | 0 | 2 | 0 | 0 | **458** | 0 | 0 | 17 | 12 |
| LAM | 0 | 0 | 22 | 0 | 10 | 1 | **1145** | 0 | 272 | 40 |
| MMI | 0 | 0 | 1 | 0 | 0 | 0 | 0 | **47** | 0 | 1 |
| BAR | 0 | 0 | 0 | 0 | 4 | 42 | 154 | 0 | **687** | 28 |
| IAS | 0 | 0 | 0 | 0 | 0 | 176 | 10 | 0 | 74 | **740** |
| UA (%) | 93.5±7.2 | 75.0±13.6 | 81.9±1.8 | 84.7±5.1 | 81.0±3.6 | 93.7±2.2 | 76.8±2.1 | 95.9±5.6 | 75.1±2.8 | 74.0±2.7 |
| PA (%) | 100 | 100 | 95.1±1.0 | 47.1±5.7 | 82.0±2.9 | 62.7±2.7 | 78.3±2.0 | 100 | 62.2±5.9 | 90.4±1.9 |
| OA (%) | | | | | 79.3±1.0 | | | | | |

To further evaluate the CALC-2020 mapping performance over space, we divided the entire study area into 50 km × 50 km grids. Then the classification confidence in each grid was computed as the proportion of correctly classified validation sample points (**Figure 4**). Overall, we estimated that the CALC-2020 classification confidence is 0.795 (±0.323, one standard deviation). Among 2,037 grids that have at least one validation sample record, 1,406 (60.9%) show confidence levels higher than 0.75. Theses grids are representative over space, by county, and by continents. The dominance of high confidence grids mirrors small percentages held by those having low confidence levels (less than 0.25). Spatially, hotspots of large classification uncertainty were commonly detected in regions with sparse validation sample distribution (**Figure S3**), such as Greenland periphery, central Siberia and American Arctic Archipelago.

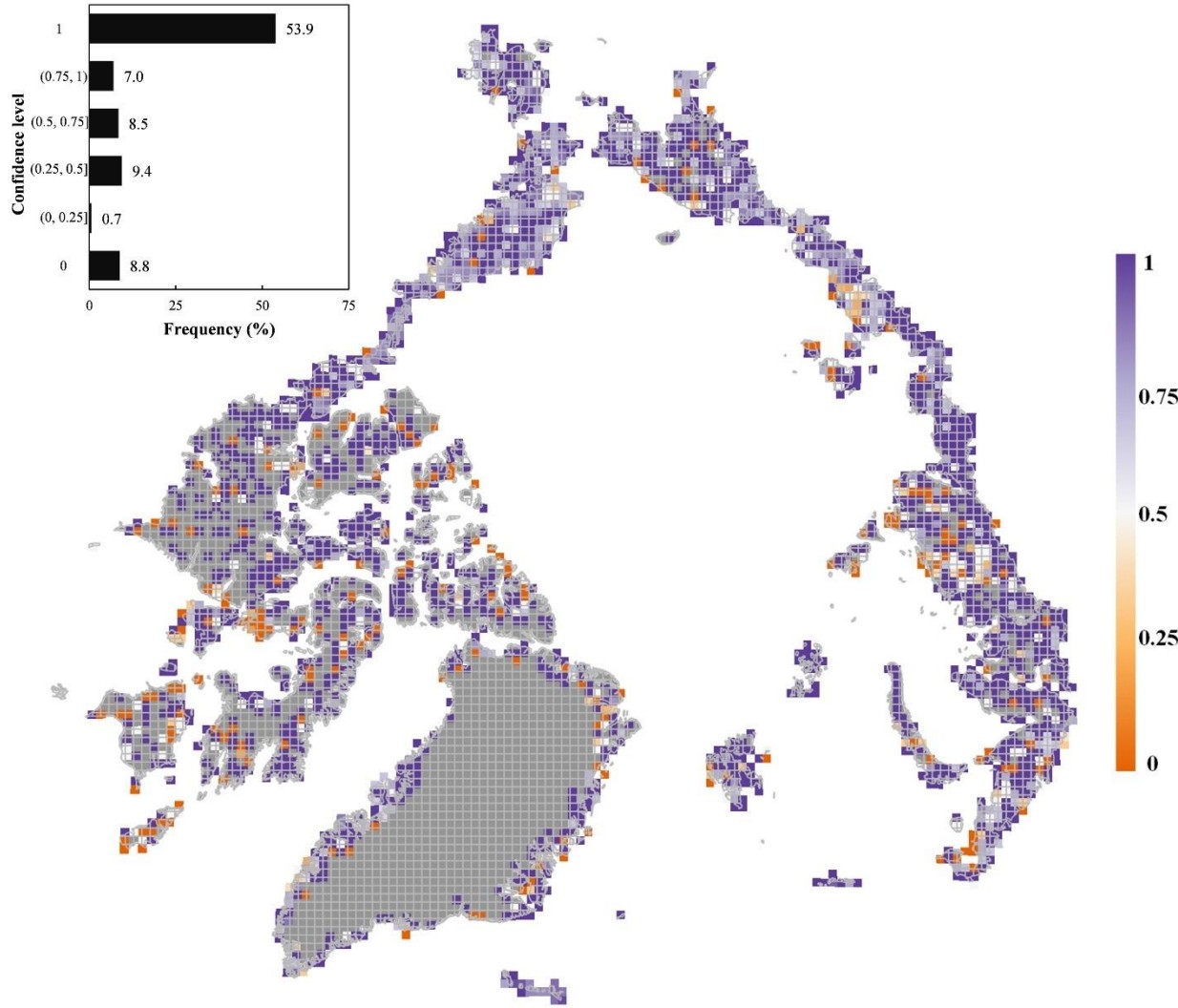

**Figure 4: Map of CALC-2020 classification confidence calculated at a 50 km × 50 km tile scale. The grey grid denotes no validation sample point distribution. The bar plot shows the statistical distribution of six broad classification confidence intervals.**

### 3.1.2 Land cover mapping performance in field and flux tower sites

**Figure 5** displays the alluvial diagram of performance evaluation for the CALC-2020 product in field and flux tower sites.
Overall, our estimation is reasonably consistent with the in-situ reports, and outperforms three widely used global land cover products (**Figure S5**). There are 37 out of 55 sites showing consistent classification results, with OA value being equal to 67.3%. When the analysis is broken down into the biome level, the biggest error source comes from confusion among different vegetation types. In particular, there are six wetland sites and three shrubland tundra sites that were mistakenly identified as graminoid tundra. These discrepancies reflect the complex suite of factors that can obscure the correct identification of Arctic biomes. For example, some wetland vegetation species are morphologically similar to gramineous plants, thus limiting the classification accuracy (Magnússon et al., 2021). The relatively poor mapping performance of Arctic vegetations could be also attributed to the short growing season (typically ranging from 50 to 60 days), in which satellite coverage is commonly spatially and temporally uneven (Beamish et al., 2020). Considerable misclassifications were also observed in some sites dominated by man-made impervious surfaces, suggesting the technical challenge of capturing small-scale artificial imperviousness using the CALC-2020 map. It is important to point out that the field and flux tower sites used in the present work are not evenly distributed over space and across biomes. Some land cover types (e.g., forest and lichen/moss) have very limited sites after data screening, making them less representative for mapping performance evaluation. This situation would be improved as more ground and near-surface reference data become available in the future (Richardson et al., 2018; Pastorello et al., 2020).

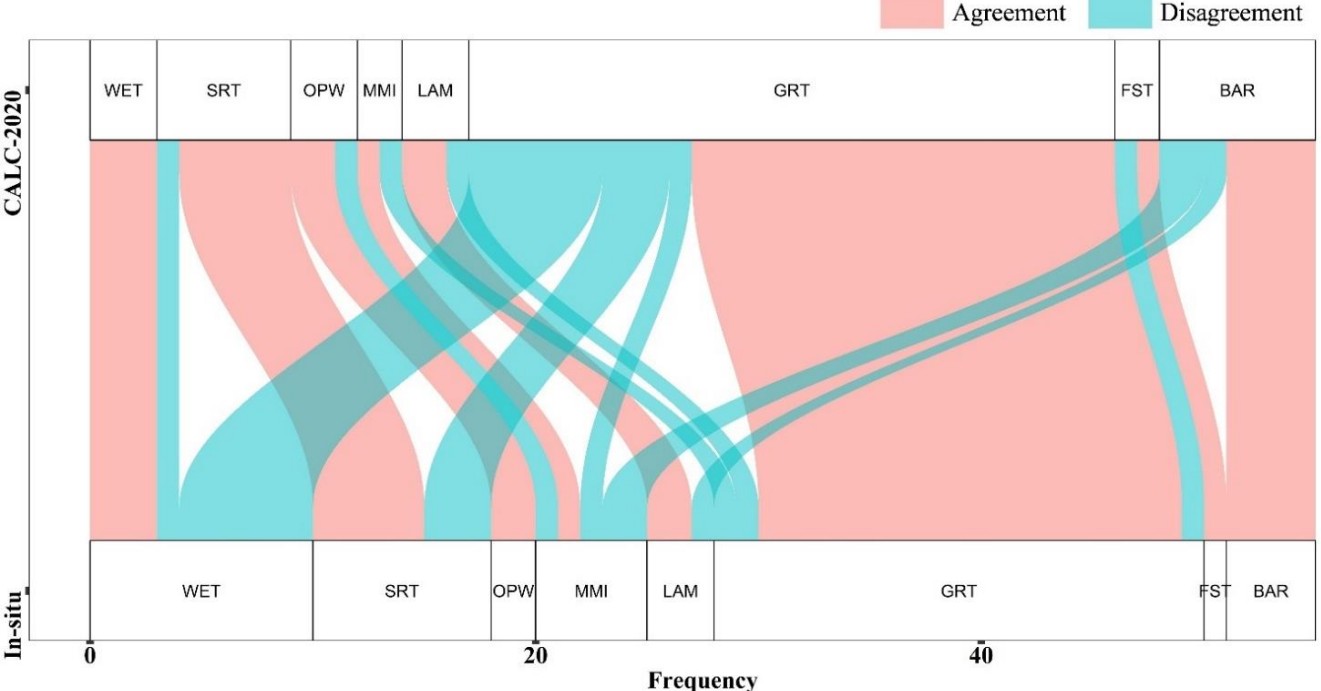

**Figure 5: CALC-2020 map performance in 55 field and flux tower sites. The block width represents the frequency (site number) identified by our estimation and in-situ reports, respectively. Land cover abbreviations are given in Table 1.**

### 3.1.3 Comparison with existing land cover products

**Figure 6** displays the spatial patterns and statistics of agreement/discrepancy between the CALC-2020 map and three global land cover products. At the circumpolar Arctic scale, classification agreements (area proportion of AG) range from 23.1% (ESRI Global Land Cover) to 45.4% (GlobeLand30) by treating our estimates as the baseline. Spatially, large AG variations were also detected across biomes and among countries. The mapped disagreements were induced by multiple factors. First and foremost, the CALC-2020 classification scheme is different from those of compared land cover products. This leads to considerable pixels identified as DS, especially for ESA WorldCover (29.7%) and ESRI Global Land Cover (56.5%) in which the graminoid tundra class is absent. Another issue that can cause the inconsistency is the difference of classification model prediction. For example, in Canadian Arctic, a latitudinal north (high)–south (low) contrast in land cover mapping agreement is evident, when comparing our map to the GlobeLand30 dataset. Such a discrepancy is primarily due to the misclassification of lichen/moss as graminoid tundra by GlobeLand30. Consistently across all countries, DD plays a minor role with very limited area occupation (less than 5%). Using the same sample-based evaluation approach applied to CALC-2020, we reported limited classification accuracies of three global land cover products for the circumpolar Arctic (**Figure 7**), with OAs ranging from 48.5% to 71.2%. In the meantime, these global-scale datasets exhibit wide PA and UA variations, implying imbalanced mapping performances across different Arctic land cover types (Liang et al., 2019).

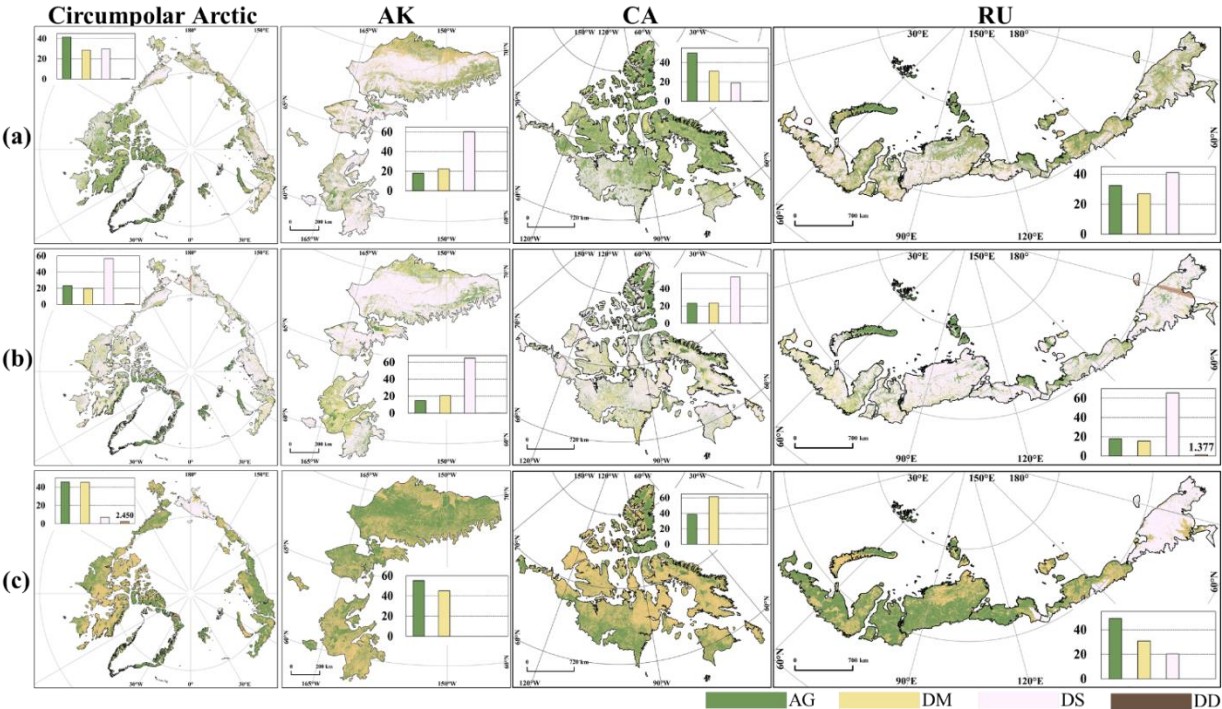

**Figure 6: Spatial distributions of the classification consistency between the CALC-2020 map and three global land cover products including the ESA WorldCover V100 (a), the ESRI Global Land Cover (b), and the GlobeLand30 V2020 (c). The bar plot in each panel shows the pixel frequency distributions (%) of the four categories: classification agreement (AG), classification disagreement due to model prediction (DM), classification disagreement due to scheme difference (DS), and classification disagreement due to data missing (DD). Statistics less than 1% are not displayed.**

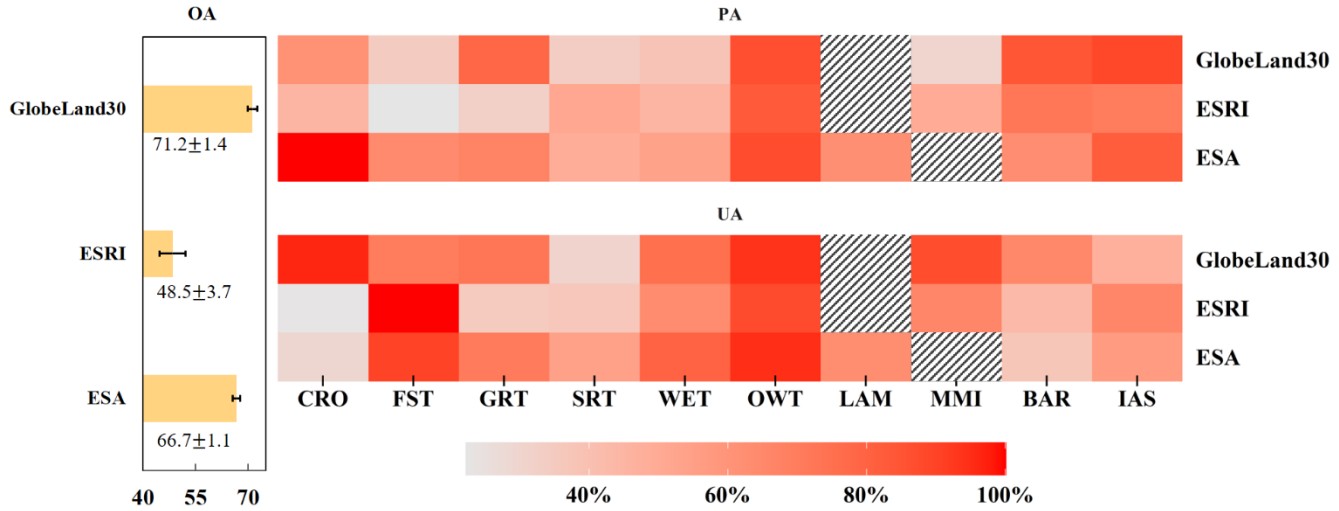

**Figure 7:** Accuracy statistics of three global land cover products for the circumpolar Arctic based on validation sample. Grass (ESA WorldCover, ESRI Global Land Cover) and wet tundra classes (GlobeLand30) are treated as equivalents of graminoid tundra and wetland, respectively. Stripped blocks represent the absence or less than 0.25% area proportion of specific class(es).

**Figure 8** further compares multiple land cover datasets by selecting four sub-regions, each of which represents one typical landscape across the terrestrial Arctic. The Google Earth image of each sub-region is also displayed for assisting mapping performance evaluation. All land cover products are displayed with their corresponding classification schemes. In Keewatin (Canada), all the products correctly capture most open water areas. Compared to other land cover datasets, the CALC-2020 map is more similar to GlobeLand30 in terms of the overall land cover composition and distribution. In North Slope (Alaska), our estimation detects the coexistence of graminoid tundra and shrub tundra, with their distributions highly related to topographic characteristics. This heterogenous land cover pattern, however, is not observed in other three products. The reasonable performance of the CALC-2020 map for North America was also confirmed by referring to two national-scale land cover products: NLCD and Land Cover of Canada, both of which exhibit high agreement of land cover distribution pattern with CALC-2020 at their level-1 classification schemes (**Figure S6**). For Russian Arctic, the largest mapping discrepancy among different mapping results is found in Yamal Peninsula, where more than half of the landmass is covered by thermokarst lakes. Our dataset is generally consistent with GlobeLand30, but providing much more spatial details. ESA WorldCover and ESRI Global Land Cover, on the other hand, show greater wetland estimates. Moreover, CALC-2020 is the only land cover product that fully depicts the distribution of man-made impervious surfaces (oil/gas deposits and traffic pavements). In Nenets, major forest clusters are correctly identified by the three datasets with a finer resolution of 10 m, including CALC-2020, ESA WorldCover, and ESRI Global Land Cover. However, the latter two products are not able to isolate graminoid plants from wetlands, both of which are clearly displayed by our mapping result. In summary, our estimations capture more subtle polar biome patterns than three global land cover products, although they were generated to depict general-purpose land cover at global scales.

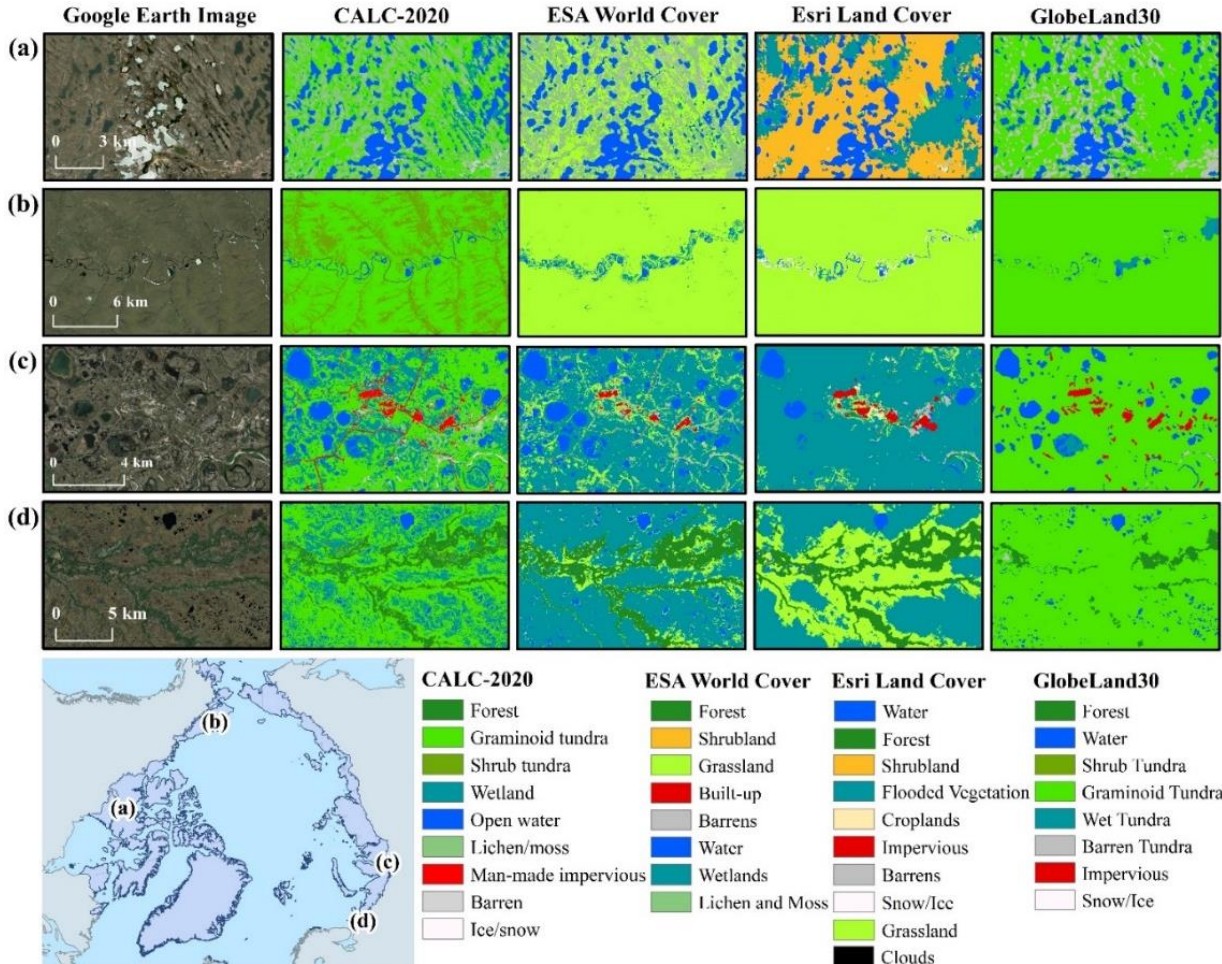

**Figure 8: Comparison of CALC-2020 mapping results with three land cover products in typical sub-regions. (a) Keewatin in Canada, centred at 65.3°N, 99.1°W. (b) North Slope in Alaska, centred at 69. 5°N, 156.0°W. (c) Yamal Peninsula in Russia, centred at 67.9°N, 75.5°E. (d) Nenets in Russia, centred at 66.6°N, 47.2°E. © Google Earth.**

**3.2 Spatial patterns and composition of circumpolar Arctic land cover**

The CALC-2020 product provides the first spatially continuous map of circumpolar Arctic land cover at 10 m resolution (**Figure 9a**). Based on this map, we calculated the distribution densities of all land cover types at the 1°×1° tile scale (**Figure 9b~k**), as well as their total area statistics throughout the terrestrial Arctic using the error-adjusted area estimation strategy (Olofsson et al., 2014) (**Figure 9l**). Among all land cover classes, the graminoid tundra occupies the largest Arctic land area

(1473,011$\pm$33972 km$^2$, 24.9%), closely followed by the lichen/moss class (1368,916$\pm$23115 km$^2$, 23.2%). In contrast, croplands and man-made impervious surfaces play a very minor role with limited area occupation (less than 1,000 km$^2$). Spatially, clustered hotspots of forest and shrub tundra are only found in Alaska and Southern Nenets in Russia. Stress-tolerant biomes (i.e., graminoid tundra and lichen/moss), on the other hand, occupy the most parts of the terrestrial Arctic, although the latter exhibits a northward distribution shift. For Arctic wetlands, a latitudinal north (less)–south (more) contrast in

fractional coverage is evident, which closely corresponds to the distribution of open water areas. Conversely, barren and ice/snow coverages are more frequently detected in middle to high Arctic regions, such as Greenland periphery, Svalbard archipelago, and Canadian Arctic Archipelago. Our map also provides observational evidence of human action on Arctic landscapes, primarily via man-made imperviousness encroachment. This is the result of persistent disturbances from industrial infrastructure development and traffic pavement (Bartsch et al., 2021; Xu et al., 2022b).

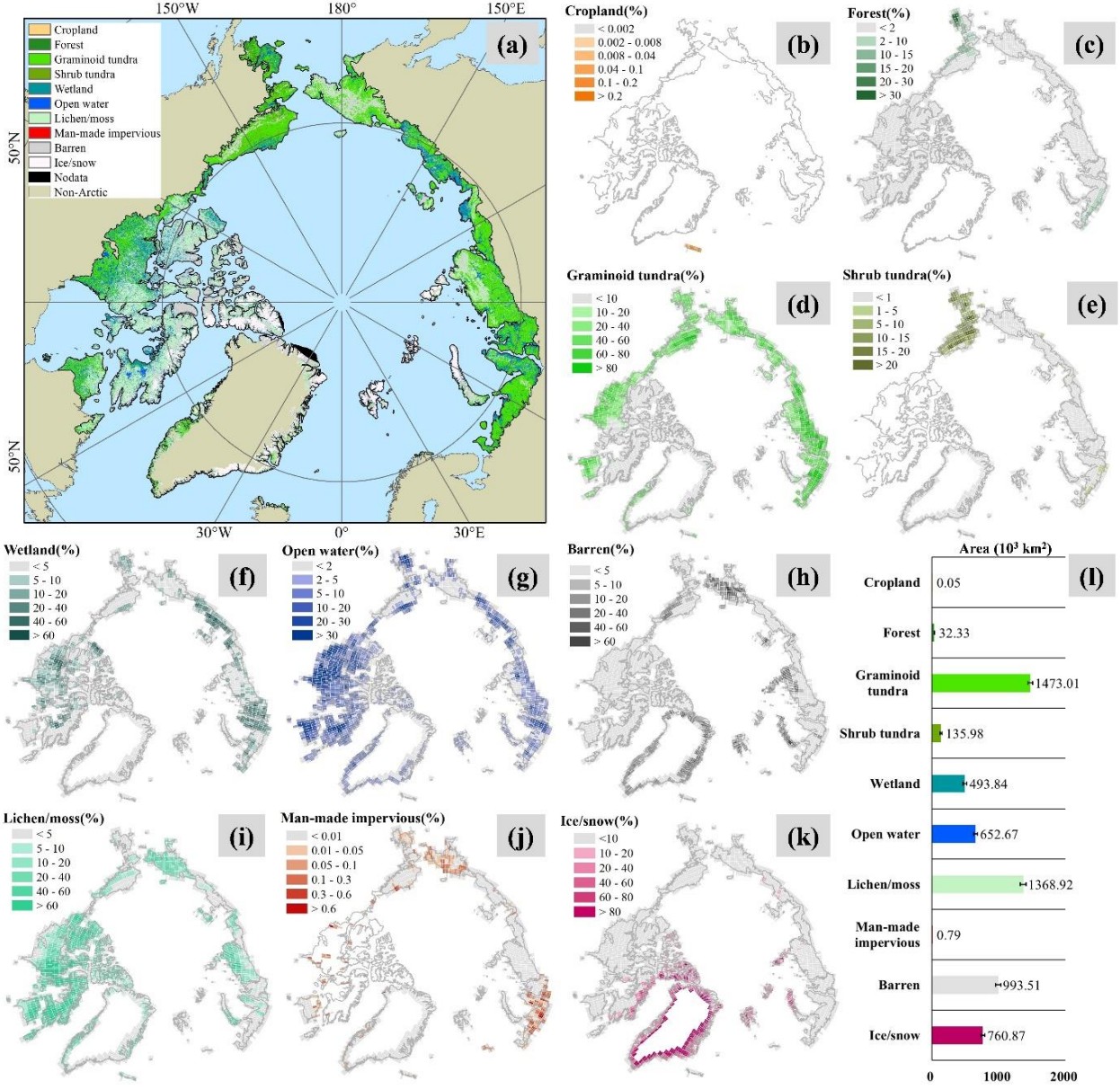

**Figure 9: Circumpolar Arctic land cover estimated by the CALC-2020 map. (a) Land cover distribution at the 10 m pixel scale. (b)~(k) display the biome-specific area proportion in each 1°×1° land tile. (l) shows area statistics of all land cover types across the terrestrial Arctic (unit of 10³ km²). Error bars represent 95% confidence intervals.**

By disaggregating land cover composition at multiple scales, The CALC-2020 map offers mechanistic insights into potential
controls on biome distribution within the Arctic (**Figure 10**). We found that land cover compositions are unevenly distributed
across countries (**Figure 10a**). The largest area proportion of vegetated coverage occurs in Alaska (85.8%), which is
comparable to Russia (80.4%). In contrast, Canada and three Nordic countries/regions (Greenland, Iceland and Norway) have
more non-vegetated areas covered by open water bodies, barren lands and snow/ice grounds. Consistently among all countries,
stress-tolerant biomes (graminoid tundra and lichen/moss) play a more prominent role than the other vegetated classes. In
addition, Alaska is the only statistical unit with over 20% landmass occupied by woody plants (shrub tundra and forest).

To investigate the climate effect on land cover composition, we further divided the entire terrestrial Arctic into five bioclimate
zones (Walker et al., 2005; Raynolds et al., 2019) defined by summer warmth index (SWI, the sum of monthly mean air
temperatures greater than 0°C) (Jia et al., 2003). We found a clear transition in land cover composition from bioclimate zone
A to E (**Figure 10b**). The overall area proportion loss in snow/ice and bare land mirrors vegetation cover gain, together
confirming that warm conditions are generally optimal for Arctic plant growth (Keenan and Riley, 2018). Within various
vegetation classes, graminoid plants exhibit the largest area increase as SWI gradually increases.

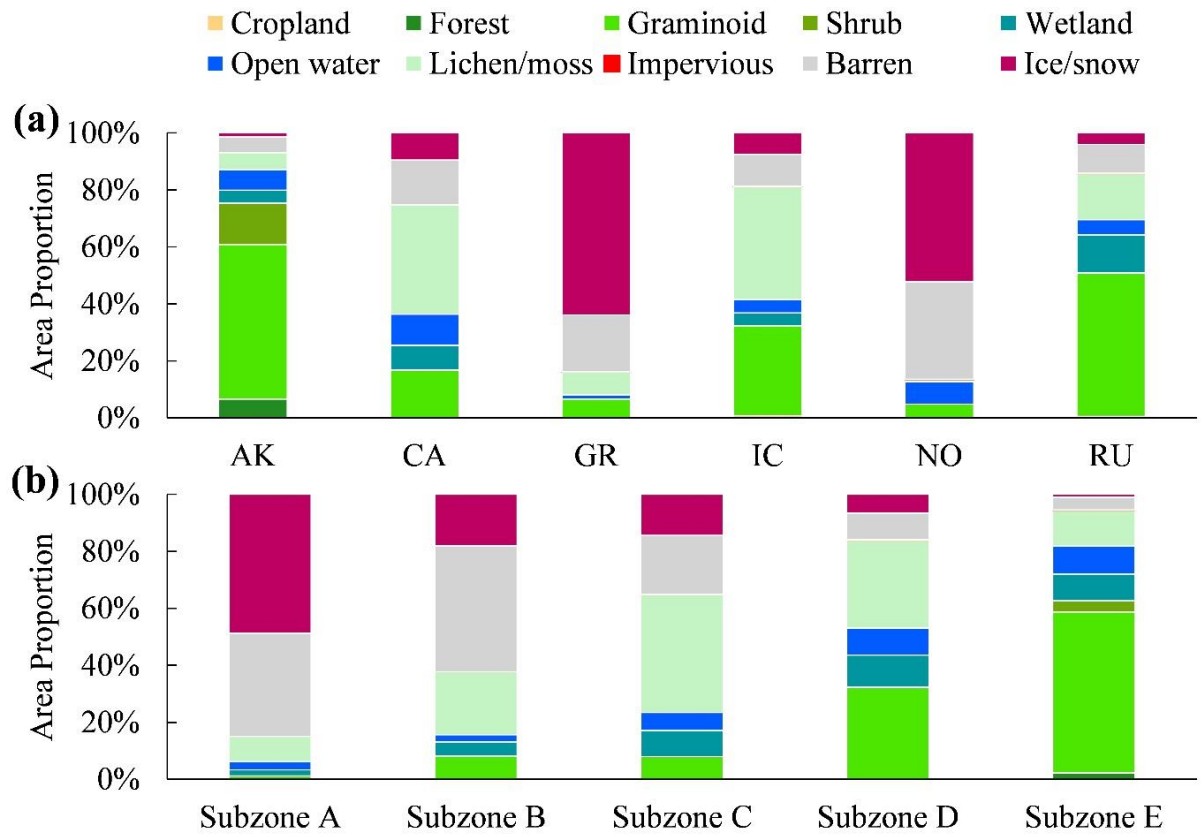

**Figure 10: Circumpolar Arctic land cover composition disaggregated at country (a) and bioclimate zone (b) scales using the map-based area estimation strategy. Subzone A~E correspond to SWI sections: SWI<6, 6≤SWI<9, 9≤SWI<12, 12≤SWI<20, and**
**SWI≥20 (unit in °C). Country abbreviations are given in Section 2.1.**

### 3.3 Methodological and scientific implications

Fine resolution mapping of circumpolar Arctic land cover is a challenging task from almost every aspect of satellite remote sensing. Our study highlighted the necessity of including both active and passive Earth observations to create a seamless land cover map across the entire terrestrial Arctic (Bartsch et al., 2016). We observed high cloud contamination (clear Sentinel-2 observation percentage less than 40%) in over half of the Arctic landmass (**Figure 2**), where Sentinel-1 SAR images can be particularly helpful to improve spatial integrity of the resultant map. The utilization of multi-source features also benefits distinguishing land cover types that are difficult to classify from the spectral domain alone. **Figure 11** displays the feature importance, quantified by total decrease in Gini impurity index over all trees in the RFC model. Consistently across all countries, topography is the most helpful feature domain, which is in line with previous studies and supports the idea that, at high latitudes of the Northern Hemisphere, the land cover composition and distribution are highly subject to terrain conditions (Walker et al., 2005; Jin et al., 2017). Meanwhile, topographic features show large importance variations among and within countries, reflecting the existence of local ecological forces that can affect the land cover pattern. In this study, we found that Sentinel-1 derived features are more effective than those from Sentinel-2. This result highlights the necessity of including all-weather capable SAR data for identifying circumpolar Arctic land surface information (Lönnqvist et al., 2010; Zhang et al., 2020). Another factor probably obscuring spectral features' usage is that some of them only have substantial impacts on certain land covers (Friedl et al., 2010; Huang et al., 2022). For example, phenometrics can benefit classifying different vegetation types, yet are less effective when distinguish man-made imperviousness from barren land.

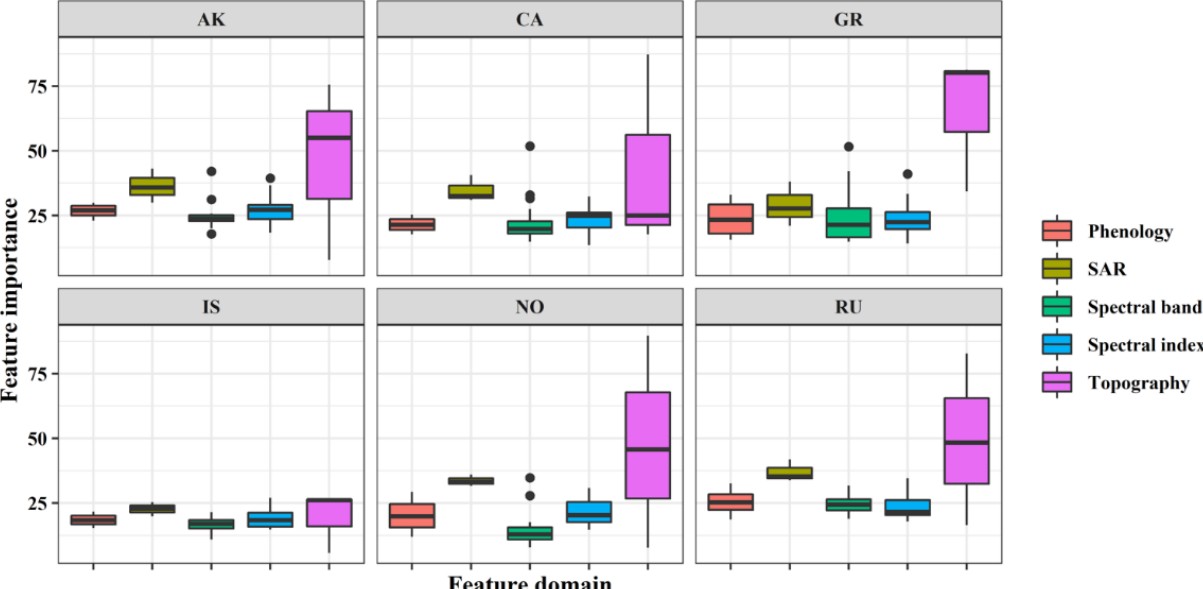

**Figure 11: Box plots showing the contribution variation among different feature domains at the country level. The feature importance is measured by total decrease in Gini impurity index over all trees in the RFC model.**

The land cover legend should reflect the information content of the study area to be interpreted (Wulder et al., 2018; Song et al., 2018; Zhang et al., 2021a). Currently, debate still exists on the determination of classification scheme system over the terrestrial Arctic (Bartsch et al., 2016). For example, the widely used IGBP classification system (Friedl et al., 2010; Loveland and Belward, 1997) contains 17 categories, but very few of them appear at the northern high latitudes (Liang et al., 2019). By contrast, some well-known polar biomes (e.g., graminoid tundra and lichen/moss) are absent in existing global land cover products (**Figure 8**), making the description of complex landscape over-simplified. In the present study, we designed a ten-category classification scheme that is generally consistent with the Circumpolar Arctic Vegetation Map (CAVM) (Walker et al., 2005; Raynolds et al., 2019), thus carrying the potential for discriminating Arctic plant communities at a fine spatial resolution. However, it is worth noting that some biomes exhibit very strong intra-class variability in physiognomy, which makes the proposed classification scheme less desirable. For example, a short shrubland tundra environment is a fundamentally different ecosystem from a tall one and this influences a vast array of biotic and abiotic processes (Walker et al., 2005). With updates of more multi-source Earth observation data in the future (e.g., vegetation height), the development of Version 2 CALC product will become a possible topic, which is expected to have a hierarchical classification scheme and improved mapping performances for some specific biomes (e.g., shrub tundra).

Generating reliable training and validation data has always been a critical constraint on land cover mapping applications. Traditionally, training sample can be either collected from field surveys (Gong et al., 2020b) or interpreted from remotely sensed images (Liu et al., 2019; Brown et al., 2020; Potapov et al., 2022). These approaches, however, are laborious or even unrealistic in remote and inaccessible areas, such as the Arctic. Alternatively, several studies demonstrated the potential of deriving training sample from pre-existing knowledge (Gray and Song, 2013; Hakkenberg et al., 2020). Building on this premise, our study developed a "ready for use" training sample by leveraging the FAST library and pre-existing land cover datasets for supervised classification model development. This sample migration strategy can be applied to various ecological zones, and particularly useful in ecoregions where land cover reference data are not available. In addition to the sample-based validation, we also evaluated the CALC-2020 product with other two assessment data sources: in-situ records and contemporary land cover products. With precisely known location and relatively homogeneous biome footprint, near-ground site networks (e.g., FLUXNET, PhenoCam) provide the most unbiased information for land cover mapping accuracy assessment (Gong, 2008). Unfortunately, these networks are currently sparse in the Arctic, and the site number of some land cover types is very limited (**Table S2**, **Figure 5**), making them less representative for Pan-Arctic applications. Comparing the CALC-2020 map with existing global land cover products is complementary to the sample/in-situ evaluation in characterizing pixel-level agreement, and we found multiple factors (DM, DS, and DD) that can cause the mapping inconsistency (**Table 2, Figure 6**). Along with these factors, DM represents the only common algorithm mechanism across all compared products. Rather than adopting a universal predictive framework for the entire study area, the land cover class prediction of the CALC-2020 map was derived from locally adaptive (country-specific) RFC models, so the unfavourable impacts incurred by geographical variability can be largely reduced (Zhang et al., 2021a; Huang et al., 2022). However, the use of country-specific

RFC models inevitably caused the spatial discontinuity issue in borderlands of neighbouring countries, which requires further methodological improvements in the future.

Arctic ecosystem function is highly dependent on land cover composition and distribution, yet both of them remain poorly understood (A'Campo et al., 2021; Wang and Friedl, 2019; Beamish et al., 2020). We expect that the CALC-2020 product will help fill the scientific gap by providing the most recent circumpolar biophysical conditions in the Northern Hemisphere. For

example, accurate land cover data are key inputs for projecting biogeochemical cycles under current and future scenarios, thus guiding local, national, and global efforts of climate change mitigation (Horvath et al., 2021; Zhang et al., 2021b). Previous studies have found a poleward movement of the Arctic treeline—northernmost edge of the habitat where trees are capable of growing. Our estimation that circumpolar Arctic forest cover reaches approximately 32,000 km$^2$ suggests woody encroachment as the climate warms, and is thus consistent with the forest growth trend (Harsch et al., 2009; Rees et al., 2020). However, our

results differ from previous studies by identifying the subtle distribution pattern of trees with a much finer spatial resolution (**Figure 12**). Finally, using the CALC-2020 product as the baseline, along with decades of satellite observation wealth, circumpolar land cover change monitoring can be possibly crafted, which will advance our understanding of a continuously changing Arctic and its global environmental impacts.

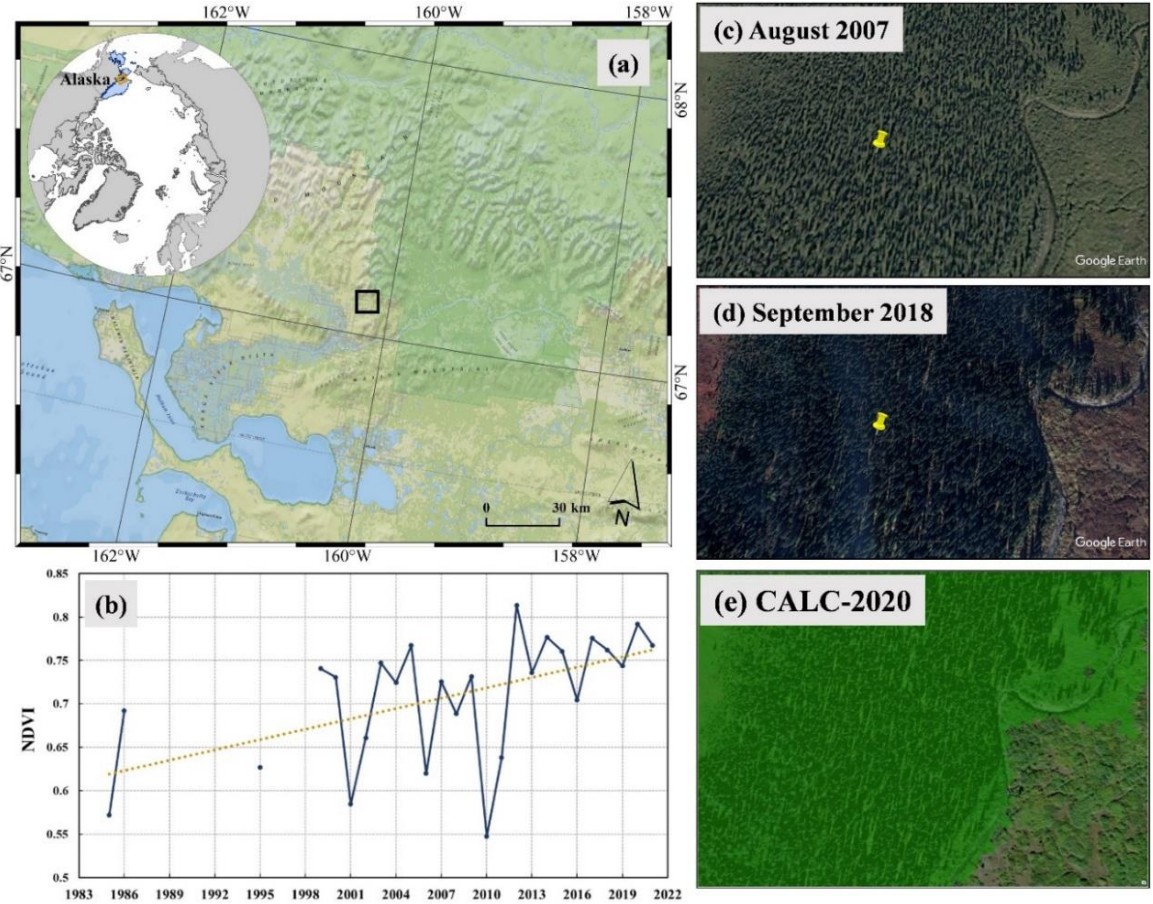

**Figure 12: Satellite observed afforestation in the Arctic. (a) Location of afforestation centred at 67.1°N, 160.2°W; (b) annual median Landsat NDVI time series from 1984 to 2021; (c)~(d) comparison of two VHR images acquired in 2007 and 2018, respectively (© Google Earth); (e) forest extent derived from the CALC-2020 map. The topographic map was generated using ESRI basemap layer product.**

## 4 Data availability

The CALC-2020 product generated in this paper is available on Science Data Bank: http://cstr.cn/31253.11.sciencedb.01869 (Xu et al., 2022a). Across the entire Arctic, the CALC-2020 product consists of six files in GeoTIFF format with a 10 m spatial resolution (EPSG: 6931). Each land cover map file is named based on the following rule: "CALC-2020-X.tif". The "X" in the file name represents its mapping country (AK, CA, GR, IC, NO, RU). Country abbreviations are given in **Section 2.1** of this paper. The valid values for circumpolar Arctic land cover types are 1~10. The CALC-2020 product was generated on the GEE platform using the JavaScript language developed by the authors. All other data used in this study are available from the corresponding authors upon reasonable request.

## 5 Conclusions

A thorough understanding of the Arctic terrestrial surface requires information about both the composition and the distribution of land cover. In this study, we developed a circumpolar Arctic land cover product for circa 2020 using fine resolution multi-source remote sensing data. Based on a "ready for use" training sample set derived from multiple sources, the CALC-2020 map was generated through a locally adaptive machine-learning classification procedure. Accuracy assessments reveal the reliability of CALC-2020, as well as its representativeness for characterizing the Arctic ecosystem in ways that are not well represented by pre-existing products. According to our estimation, the graminoid tundra and lichen/moss occupy the largest Arctic land area, and the latitudinal shift of land cover composition is generally consistent with the SWI gradient profile. Our mapping results also offer the evidence of woody encroachment, especially in Alaska and Southern Nenets, Russia. We concluded that the new CALC-2020 map can be used to augment the modelling of both biotic and abiotic processes, thus enlightening innovative Arctic management.

## Author contributions

Chong Liu, Caixia Liu, and Huabing Huang carried out the analysis and wrote the manuscript; Xiaoqing Xu and Xuejie Feng helped with the data processing and the accuracy assessment; Xiao Cheng conceived the study. All authors helped revise the manuscript.

**Competing interests**

The authors declare that they have no conflict of interest.

**Financial support**

The research was supported by the Open Research Program of the International Research Center of Big Data for Sustainable Development Goals (CBAS2022ORP04), the National Key R&D Program of China (2018YFC1407103, 2019YFC1509105, 2019YFC1509104), Guangdong Natural Science Foundation (2022A1515010924), and the Major Key Project of PCL.

**Acknowledgments**

We acknowledge the Google Earth Engine platform, which makes circumpolar Arctic-scale geospatial mapping and analysis 490 possible at fine spatial resolutions. We are also grateful to all data providers that have been used in this study. The authors would like to thank the topical editor and two anonymous reviewers for their constructive and insightful comments on an earlier draft of this paper.

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
