# Peer review of "CALC-2020: a new baseline land cover map at 10 m resolution for the circumpolar Arctic"

_Earth System Science Data, 2022_

## Author Response (AR1)

Chong Liu
School of Geospatial Engineering and Science
Sun Yat-Sen University

Topical Editor
Earth System Science Data

November 16, 2022

Dear Editor,

We very appreciate your providing us with the reviewers' comments. We have addressed and responded to the comments point by point, and revised our manuscript accordingly.

Major changes are summarized as follows:

1)  We revised the **Introduction** by clarifying the necessity of developing CALC-2020 and highlighted the reasons why using multi-source remote sensing data. See **Section 1**.

2)  We added detailed descriptions of generating the training sample, in particular rewriting the first paragraph of **Section 2.3.1**.

3)  We developed a completely independent validation sample, and employed it assessing the accuracy and uncertainty of our resultant land cover map, based on good practices by Olofsson et al. (2014). See **Section 2.4, 3.1.1**, and updated **Table 3**, **Figure 4**.

4)  We clarified the necessity comparing CALC-2020 to three widely used global land cover products, and added a new table explaining the meanings of AG, DM, DS, and DD. See **Section 2.4** and **Table 2**.

5)  We added a new section (**Section 2.5**) for explaining our strategies of land cover area estimation.

6)  The author affiliations were updated in the revised manuscript.

Thank you for your time and consideration of the revised manuscript. We look forward to hearing from you.

Sincerely,

Chong Liu
Contact author

**CALC-2020: a new baseline land cover map at 10 m resolution for the circumpolar Arctic**

Chong Liu, Xiaoqing Xu, Xuejie Feng, Xiao Cheng,
Caixia Liu, and Huabing Huang

**Responses to Referee #1**

**November 16 2022**

We very appreciate the topical editor and two reviewers for constructive comments, which are very helpful for the improvement of the paper. We have seriously addressed these comments and revised the paper accordingly. The following are the summary of major changes and our point-by-point responses, indicated in blue.

**General comments**

The authors created a new 10-m land cover map for the circumpolar Arctic based on the combined use of Sentinel-2, Sentinel-2, and DEM and multiple-source training samples. It is challenging to map land cover over the Arctic due to the limited number of snow/ice-free observations. Yet, the current manuscript does not well demonstrate the land cover product is promising, particularly for the validation and the comparison. I have also lots of questions regarding the method. Please see my detailed comments below.

**Response**: We would like to thank the reviewer for handling our manuscript and pointing out key issues that help significantly improve the research and the quality of this work. We revised the manuscript according to these comments, and below we provided our detailed responses to the points raised.

**Major comments**

All the samples filtered by the SOM may be helpful to improve the classification, but it will select similar samples for each category. The validation data, with the same resources as training data, will overestimate the accuracy of the land cover map. The samples from flux tower sites may be OK, because of the real independence to the training samples, but the number is too limited, and serval important land cover types are lacking, such as cropland, forest, snow/ice (Table S2). Here I highly recommend the authors create new and independent validation samples stratified on the Arctic land cover map (see Olofsson et al. 2014, RSE, good practices).

**Response**: We appreciate this thoughtful comment. The primary purpose of using SOM in this study was to refine the preliminary training sample derived from different sources. As the reviewer mentioned, the SOM filtered sample may suffer from "over-similarity", which makes resultant sample less informative. We did recognize this issue, and have included a necessary procedure when implementing SOM algorithm. In this procedure, for each land cover type, we divided the sample into 100 clusters, thus the inter-cluster difference reflects the intra-class variability. We clearly mentioned this in the revised manuscript (**Page 8**, **Lines 171-172**).

We agree that all the validation statistics reported in the paper should be solid and convincing. For this purpose, we followed the reviewer's suggestion of creating a completely independent validation sample using good practices by Olofsson et al. (2014). We used CALC-2020 map itself as the stratification of study area, and set the validation sample size to 6,513 by specifying a target standard error for overall accuracy (OA) of 0.005. We allocated 40~1872 stratified sample units for each class (46, 40, 1,832, 189, 463, 489,1,490, 49, 915 and 1,000 points for cropland, forest, graminoid tundra, shrub tundra, wetland, open water, lichen/moss, man-made impervious, barren, and ice/snow, respectively), and calculated error metrics including user's and producer's accuracies (UA and PA), along with estimates of associated 95% confidence intervals. The reference class label for each sampled pixel was identified based on expert interpretation of cloud-free Sentinel-2 images and Google Earth VHR imagery data, as available. Sample pixels with disagreement among experts were subsequently revisited until a consensus was reached. Here we offer **Figure R1** and **Table R1**, showing the newly developed validation sample distribution and the error matrix, respectively. Based on the newly-built validation sample, the OA of CALC-2020 map is 79.3 ± 1.0% (95% confidence interval). Please find detail descriptions on validation sample collection (**Page 9-10**, **Lines 216-223**) and analysis (**Page 11-12**, **Lines 253-271**) in the revised manuscript.

Given the capacity of depicting real land surface environments, in situ data provides the most valuable reference for land cover mapping accuracy assessment. Unfortunately, these sites are so far scarce over space and biomes within our study area. This situation would be improved as more high-quality site data become available in the future. We point out this limitation in the revised manuscript (**Page 13**, **Lines 288-290**).

[Figure]

**Figure R1** Map showing locations of newly built validation sample used in the revised manuscript.

**Table R1**: Error matrix of ten mapped classes with cell entries filled by sample number. Reference classes are in columns.

|        | CRO | FST | GRT  | SRT | WET | OWT | LAM  | MMI | BAR | IAS | Total |
|--------|-----|-----|------|-----|-----|-----|------|-----|-----|-----|-------|
| **CRO** | 43 | 0 | 3 | 0 | 0 | 0 | 0 | 0 | 0 | 0 | 46 |
| **FST** | 0 | 30 | 2 | 8 | 0 | 0 | 0 | 0 | 0 | 0 | 40 |
| **GRT** | 0 | 0 | 1501 | 62 | 81 | 28 | 125 | 0 | 35 | 0 | 1832 |
| **SRT** | 0 | 0 | 25 | 160 | 1 | 0 | 2 | 0 | 1 | 0 | 189 |
| **WET** | 0 | 0 | 35 | 5 | 375 | 14 | 30 | 0 | 4 | 0 | 463 |
| **OWT** | 0 | 0 | 2 | 0 | 0 | 458 | 0 | 0 | 17 | 12 | 489 |
| **LAM** | 0 | 0 | 22 | 0 | 10 | 1 | 1145 | 0 | 272 | 40 | 1490 |
| **MMI** | 0 | 0 | 1 | 0 | 0 | 0 | 0 | 47 | 0 | 1 | 49 |
| **BAR** | 0 | 0 | 0 | 0 | 4 | 42 | 154 | 0 | 687 | 28 | 915 |
| **IAS** | 0 | 0 | 0 | 0 | 0 | 176 | 10 | 0 | 74 | 740 | 1000 |
| **Total** | 43 | 30 | 1591 | 235 | 471 | 719 | 1466 | 47 | 1090 | 821 | **6513** |

*Reference: Olofsson, P., Foody, G. M., Herold, M., Stehman, S. V., Woodcock, C. E., and Wulder, M. A.: Good practices for estimating area and assessing accuracy of land change, Remote Sens. Environ., 148, 42–57, 2014.*

The description of the sample is not clear, i.e., how many samples were in the study area (L 133), how many changed pixels were removed (L 134), how did the authors determine the land over type for the pixels within the 90-m buffer (L 135), what was the strategy (L 139), etc. The strategy has a 40% maximum proportion and a 3% minimum proportion during the training data selection. How did the authors apply the strategy to "each" used land cover product, such as NLCD, Canada land cover, and GlobeLand30? This strategy was originally designed for selecting the training samples, that will be directly used to train the final random forest model; however, when the lichen/moss class was interpreted, how did the authors know its proportion? By the way, how did the authors immigrate the category of the existing products to Table 1, which are so different?

**Response**: Thank you for this comment. In the revised manuscript, we have re-organized the entire paragraph by clearly introducing the three data sources of preliminary training sample, how they were interpreted, and their exact numbers. Major modifications are as follows:

1) For sample derived from FAST, we pointed out that FAST contains 91,619 sample locations at the planetary scale, and a total of 14,579 sample records reserved in this study after spatial and temporal screening. For each FAST sample location, we created a $90 \times 90$ m square buffer, in which the land cover labels of all pixels were acquired based on Seninel-2 SCL band layer (offering per-pixel preliminary class label with an unsupervised approach). A FAST sample record was preserved only when it represented the dominant land cover type within its buffer area (i.e., greater than 50% area proportion). Please see **Page 7**, **Lines 136-144** in the revised manuscript.

2) For sample derived from pre-existing land cover products (NLCD 2016 for Alaska, Land Cover of Canada 2015 for Canadian Arctic, and GlobeLand30 V2020 for the rest terrestrial Arctic countries), we first spatially incorporated them into one single map layer by unifying their classification schemes into the CALC-2020 legend based on prior knowledge. For example, wet tundra in GlobeLand30 can be equivalent to wetland, while grassland/herbaceous in NLCD 2016 is alternative to graminoid tundra, due to their similar definitions. With the re-classified reference land cover layer, sample extraction was performed by using a stratified random sampling strategy. To ensure sample representativeness, we randomly collected 12,000 points for each class, except for forest (2,000 points), shrub tundra (5,000 points) and cropland (200 points) because of their limited area occupations. After removing points that failed the double check by senior interpreters, there remained 64,133 preliminary training sample points derived from existing land cover products. Please see **Page 7**, **Lines 144-155** in the revised manuscript.

3) For lichen/moss sample interpreted from VHR images, its sample size was set 5,000 to balance sampling representativeness and interpretation workload. We kept only well-interpreted points with high-level confidence, which eventually led to 4,913 preliminary training sample points for lichen/moss. Please see **Page 7**, **Lines 155-161** in the revised manuscript.

Here we show **Figure R2**, which displays the distribution of the preliminary training sample derived from FAST, pre-existing land cover products, and reference VHR imagery, respectively.

[Figure]

**Figure R2** Map showing preliminary training sample migrated from FAST (a), extracted from pre-existing land cover products (b), and interpreted from VHR imagery data (c).

The approach of comparing the new map to existing land products is not objective. For example, why did the authors focus on the agreement between the new map and the existing products, rather than focusing on the accuracy? How did the authors align the different definitions of land cover between products? What are the means for DM, DS, and DD? What this comparison analysis delivered to audiences is a huge disagreement among the maps, and so what? How about the accuracy of each product? At least, the authors can provide the accuracy against the samples from flux tower sites.

**Response**: Thanks for raising the issue on clarification. We agree that the inter-product comparison between CALC-2020 and existing products cannot directly measure the accuracy of our estimation. Instead, it provides an overall insight of pixel-level agreement. Previous studies suggested that pixel by pixel comparison is necessary for large area land cover mapping evaluation, due to its complementarity to sample based/in situ strategies (Gong et al., 2020; Friedl et al., 2022). We clarified this point in the revised manuscript (**Page 10**, **Lines 232-234**).

Classification scheme discrepancy is the reason that blocks the potential of directly calculating error metrics of compared land cover maps for the whole Arctic. For example, neither ESRI nor GlobeLand30 includes the lichen/moss class, which will lead to biased accuracy assessments. As an alternative, we focused on per-pixel agreement, by dividing comparing results into four categories: agreement (AG), disagreement due to model prediction (DM), disagreement due to scheme difference (DS), and disagreement due to data missing (DD).

We added a new table in the revised manuscript (**Table 2**) offering detail information of definitions of AG, DM, DS and DD (also displayed below as **Table R2**), and each category has its own meaning. AG suggests that CALC-2020 classification agrees with that of reference product. DM suggests classification difference caused by model prediction. DD suggests the land cover class information is lacked in the reference products (no data or unclassified). These three categories together reflect how well our result matches to pre-existing products. DS, on the contrary, reflects the inconsistency of used classification legends. Based on these considerations, we further analyzed and discussed the spatial patterns of inter-comparison results, as well as their implications in the revised manuscript (**Page 14**).

We agree with the reviewer's suggestion, and calculated overall accuracies of each compared land cover product with in-situ data (**Figure R3**). Please note that pre-processing was performed to transform different classification schemes to CALC-2020 legend, although it may result in biased error metrics. We treated grass class in ESA and ESRI as equivalents of graminoid tundra. These assumptions are expected to generate overestimated OAs for reference products. Even so, we still found that CALC-2020 exhibited the highest OA (67.3%), followed by GlobeLand30 (45.5%) and ESA (38.2%). ESRI land cover product, in contrast, had the lowest OA (25.5%) evaluated by in-situ data records. We have added relevant descriptions in the revised manuscript (**Page 13**, **Lines 277-278** and **Figure S4**).

**Table R2**: Description of per-pixel level comparison categories between CALC-2020 and compared land cover products.

| Category (abbreviation) | Definition |
|---|---|
| Agreement (AG) | CALC-2020 and the compared land cover product display identical classification result |
| Disagreement due to model prediction (DM) | CALC-2020 and the compared land cover product display different classification results, both of which are included in the CALC-2020 map legend |
| Disagreement due to scheme difference (DS) | The compared land cover product displays a classification result which is not included in the CALC-2020 map legend |
| Disagreement due to data missing (DD) | Unclassified or data missing exhibited by the compared land cover product |

[Figure]

**Figure R3** Land cover mapping performance evaluation of three reference products in field and flux tower sites, measured by overall accuracy (a) and alluvial diagrams ((b)~(d)).

*Reference 1: Gong, P., Li, X., Wang, J., Bai, Y., Chen, B., Hu, T., Liu, X., Xu, B., Yang, J., Zhang, W., and Zhou, Y.: Annual maps of global artificial impervious area (GAIA) between 1985 and 2018, Remote Sens. Environ., 236, 111510, 2020.*

*Reference 2: Friedl, M. A., Woodcock, C. E., Olofsson, P., Zhu, Z., Loveland, T., Stanimirova, R., Arevalo, P., Bullock, E., Hu, K.-T., Zhang, Y., Turlej, K., Tarrio, K., McAvoy, K., Gorelick, N., Wang, J. A., Barber, C. P., and Souza, C.: Medium Spatial Resolution Mapping of Global Land Cover and Land Cover Change Across Multiple Decades From Landsat, Front. Remote Sens., 3, 2022.*

**Minor comments**

L 110: What were discarded from the Sentinel-2 QA band? Snow/ice observations were excluded?

**Response**: We added the description of Sentinel-2 data screening in the revised manuscript. Based on the QA60 band, invalid observations, including clouds, cloud shadows, and snow, were identified and masked. Please see **Page 9**, **Lines 194-195** in the revised manuscript.

L 157: The number of each category should be given as well.

**Response**: Agreed and added. The exact training sample size of each land cover type was given in the revised manuscript: After the SOM-based refinement, the final training sample includes 70,260 valid records, including 192, 2,836, 15,686, 4,794, 6,470, 11,729, 4,380, 11,445 and 12,728 points for cropland, forest, graminoid tundra, shrub tundra, wetland, open water, lichen/moss, barren, and ice/snow, respectively (**Page 8**, **Lines 173-176**).

Section 2.4.2 Man-made impervious surface mapping: As far as I understand, the impervious surface from CAMI-2020 has the highest priority, and the authors classified the other 9 land cover types only. If that, I suggest clarifying that the classification method did not include the impervious surface, like in Section 2.4.3.

**Response**: We agree with the careful check for clarifying that the classification of this work did not include the impervious surface. We have gone through the manuscript and made sure they were mentioned whenever needed. For example, in Section 2.4.3, we rephrased the first sentence as: After acquiring the distribution of man-made impervious surface, we conducted natural surface land cover mapping based on feature metrics derived from Sentinel-1, Sentinel-2 and ArcticDEM (**Page 9**, **Lines 191-192** in the revised manuscript).

**Manuscript essd-2022-224, First Revision**
**Submitted to: Earth System Science Data**

**CALC-2020: a new baseline land cover map at 10 m resolution for the circumpolar Arctic**

Chong Liu, Xiaoqing Xu, Xuejie Feng, Xiao Cheng,
Caixia Liu, and Huabing Huang

**Responses to Referee #2**

**November 16 2022**

We very appreciate the topical editor and two reviewers for constructive comments, which are very helpful for the improvement of the paper. We have seriously addressed these comments and revised the paper accordingly. The following are the summary of major changes and our point-by-point responses, indicated in blue.

**General comments**

The manuscript combined the multisource and multitemporal remote sensing imagery to develop the new circumpolar Arctic land cover product for circa 2020, containing 10 land-cover types especially for the tundra. The dataset is great and can be used to improve earth system modelling. However, I think there are several problems should be solved.

**Response**: We would like to thank the reviewer for the constructive suggestions that help significantly improve the research and the quality of this work. We revised the manuscript according to these comments, and below we provided our detailed responses to the points raised.

**Major comments**

In the second paragraph of the introduction, author stated that 'but their strength is limited by the over simplification of classification schemes' and used the MCD12Q1 to explain the limitations of global land-cover products. In my opinion, the statement might be inaccurate, the ESA CCI_LC, GlobeLand30, and three global 10 m land-cover products (ESA, ESRI and dynamicworld) can capture the land-cover information with various classification systems or spatial resolutions. Therefore, I suggested that the authors further added the necessity of developing CALC-2020 when there were so many global land-cover products.

**Response**: Thank you for this thoughtful suggestion, which is indeed true. In the revised manuscript, we added descriptions on why existing studies/products cannot fully meet the need

of precious Arctic land cover mapping from three aspects. 1) A few studies attempted to capture Arctic land cover using satellite remote sensing, with observations obtained from satellites of Landsat, SPOT, and Sentinel-2. But these studies focused mainly on small areas, being unable to provide spatially complete information for the entire terrestrial Arctic. 2) In parallel, some existing scientific programs have manifested remarkable achievements of general-purpose land cover maps at the global scale, including (part of) the Arctic region. However, these products bear with coarse spatial resolutions (100 m~1 km pixel size), hence raising the sub-pixel mixing issue. 3) Although the entire Earth surface witnessed a growing number of fine spatial resolution land cover products, most of them have systematically low accuracies in Arctic and boreal regions (Bartsch et al., 2016; Liang et al., 2019). Please see **Page 2**, **Lines 45-58** in the revised manuscript.

*Reference 1: Bartsch, A., Höfler, A., Kroisleitner, C., and Trofaier, A. M.: Land Cover Mapping in Northern High Latitude Permafrost Regions with Satellite Data: Achievements and Remaining Challenges, Remote Sens., 8, 2016.*

*Reference 2: Liang, L., Liu, Q., Liu, G., Li, H., and Huang, C.: Accuracy Evaluation and Consistency Analysis of Four Global Land Cover Products in the Arctic Region, Remote Sens., 11, 2019.*

In the third paragraph of the introduction, the reasons why authors used multi-source remote sensing data to develop the CALC-2020 map was not clear. Actually, the combination of multisource remote sensing observations has been demonstrated to improve the land-cover classification in so many studies because of importing new information. Namely, the combination of optical, SAR and terrain features was not particularly novel in this study.

**Response**: We appreciate for this valuable comment. Indeed, multi-source remote sensing is a widely used strategy for improving classification performance. Our work follows successful experiences of previous studies, with an explicit purpose of addressing Arctic land cover mapping challenges. 1) In the Arctic, the common presence of treeless tundra landscape patches gives rise to the "spectral confusion" issue that can lead to a decreased classification accuracy. 2) Severe cloud contamination and high solar zenith angles introduce uncertainties into the Arctic land cover mapping results derived from optical imagery. 3) Terrain coefficients can also facilitate the identification of Arctic biomes by incorporating environmental factors including temperature, solar radiation, and water availability. We clarified this in our revised manuscript (**Page 2-3**, **Lines 62-71**).

Authors used the FAST sample library and three land-cover products to derive the primary training samples and then used visual interpretation to collect lichen and moss, however, how to identify the graminoid tundra and shrub tundra samples, which was not explained in the method.

**Response**: We added detailed descriptions of this issue in the revised manuscript: Special care was taken to make a distinction between graminoid tundra and shrub tundra because they are

easily confused for a single season. Thus, time series images from Sentinel-1 and Sentinel-2 were used to support judgement as needed. Please see **Page 7**, **Lines 152-154** in the revised manuscript.

Authors used the SOM method and further used the threshold of 75% to remove the wrong samples, why chose this threshold? Can you give the number of training samples for each land-cover class?

**Response**: This is a good point! We gave much critical thinking and analysis for the determination of purity index threshold. In this study, the purity index threshold was set to 75% to balance sample size and sample robustness (Gong et al., 2019). An extremely high threshold (e.g., 95%) will lead to the passing failure of most clusters, thus the size of final sample set will be too small. In contrast, an extremely low threshold will enlarge the impacts of non-target classes, which will result in reduced classification accuracy. Gong et al. (2019) performed experiments on size reduction and errors in sample. Following their experiences, we also performed sensitivity tests and found relatively robust performances using thresholds no less than 75%. We added a sentence in the revised manuscript to explicitly address this issue (**Page 8**, **Lines 171-172**).

The number of training samples for each land-cover class was provided in the revised manuscript: After the SOM-based refinement, the final training sample includes 70,260 valid records, including 192, 2,836, 15,686, 4,794, 6,470, 11,729, 4,380, 11,445 and 12,728 points for cropland, forest, graminoid tundra, shrub tundra, wetland, open water, lichen/moss, barren, and ice/snow, respectively (**Page 8**, **Lines 173-176**).

*Reference: Gong, P., Liu, H., Zhang, M., Li, C., Wang, J., Huang, H., Clinton, N., Ji, L., Li, W., Bai, Y., Chen, B., Xu, B., Zhu, Z., Yuan, C., Suen, H. P., Guo, J., Xu, N., Li, W., Zhao, Y., Yang, J., Yu, C., Wang, X., Fu, H., Yu, L., Dronova, I., Hui, F., Cheng, X., Shi, X., Xiao, F., Liu, Q., and Song, L.: Stable classification with limited sample: transferring a 30-m resolution sample set collected in 2015 to mapping 10-m resolution global land cover in 2017, Sci. Bull., 64, 370–373, 2019.*

In section 2.4.3, authors mentioned that "Cloud-free Sentinel-2 observations were first interpolated in each pixel at an 8-day time step using penalized cubic smoothing splines", is it mean these cloudy or snow pixels were interpolated by cubic splines method? Can you provide the details and the results of the time-series sentinel-2 observations after interpolating?

**Response**: Agreed and modified. We have rewritten this paragraph to clarify the technical steps of phenometrics identification. Some key points are summarized as follows. 1) Invalid Sentinel-2 observations, including clouds, cloud shadows, and snow, were firstly identified and masked according to the QA60 band. 2) Cloud-free observations were then interpolated in each pixel at an 8-day time step using penalized cubic smoothing splines. 3) With the smoothed, seamless reflectance time series, we calculated NDVI at each temporal interval to depict the time (DOY) of the vegetation phenophase transitions. Please see **Page 9**, **Lines 194-202** in the

revised manuscript. Furthermore, we added a new figure (**Figure S2**) showing original S-2 observations, fitted vegetation cycle, and derived metrics (also shown below as **Figure R4**).

[Figure]

**Figure R4** An example fit of NDVI time series for the identification of phenometrics. Green points represent original cloud-free Sentinel-2 observations.

The CALC-2020 was individually produced for each country using RFC models, how to ensure the spatial continuity between the transition areas over two neighboring countries because of using different RFC models?

**Response**: Thanks for raising the issue on clarification. 1) We agree that a universal classification model is beneficial in terms of spatial continuity. Meanwhile, we respectfully argue that such a method has its own limitations, especially when applied to continental/global scale land cover mapping. In this study, the land cover composition variation among Arctic countries/regions is large (e.g., cropland only distributed in Iceland), highlighting the necessity and effectiveness of the nation-wide/local classification strategy (Radoux et al., 2014; Huang et al., 2022). 2) It should be noted that all Arctic countries/regions involved in this study are independent geographical units separated by oceans (i.e., not affected by the discontinuity problem), except for Alaska and Canada, between which a land border exists. Therefore, we provide three typical examples among others to test the spatial discontinuity issue in North America, as shown in the following figure. Overall, all examples exhibit reasonable spatial continuity, especially in two homogenous scenes (**Figure R5(a)** and **(c)**). Nevertheless, the level of spatial continuity is less desirable in a relatively heterogenous scene (**Figure R5(b)**), which requires further methodological improvements in the future. We have clearly highlighted this issue in the revised manuscript (**Page 20**, **Lines 419-423**).

[Figure]

**Figure R5** Three typical examples ((a)~(c)) showing the performance of spatial continuity within the border area between Canada and the U.S. (Alaska).

*Reference 1: Radoux, J., Lamarche, C., Van Bogaert, E., Bontemps, S., Brockmann, C., and Defourny, P.: Automated training sample extraction for global land cover mapping, Remote Sens., 6, 2014.*

*Reference 2: Huang, X., Yang, J., Wang, W., and Liu, Z.: Mapping 10-m global impervious surface area (GISA-10m) using multi-source geospatial data, Earth Syst. Sci. Data.,14, 2022.*

The comparisons between CALC-2020 with three global maps were unfair for global land-cover products, because the CALC-2020 only focused on the circumpolar Arctic while the global land-cover maps emphasized the comprehensive performance over the globe.

**Response**: We agree that the inter-product comparison cannot directly reflect the accuracy of our estimation. Instead, the comparing results provide an overall insight of pixel-level agreement, both statistically and spatially. We clearly mentioned this in the revised manuscript (**Page 10**, **Lines 232-234**). As a complementary means apart from sample/in situ level validation, the need of inter-product comparison in this study is twofold. 1) We would like to check the spatial consistency of our approach (including classification scheme) against to widely-used fine resolution land cover products covering the extent of terrestrial Arctic. 2) We would like to evaluate the mapping performance variation of our approach across different Arctic landscapes. Based on the above reasons, three existing products were used, although they were generated to depict general-purpose land cover at global scales.

In the two comparison figures of Figure 6 and 7, I doubt the difference mainly came from the different definition for the same land-cover class especially in the Figure 7.

**Response**: Thank you for this comment. As you mentioned, there exist considerable classification scheme discrepancies between CALC-2020 and existing land cover products, which makes our validation/comparison (especially at the per-pixel level) challenging. We do recognize this issue, and have included necessary procedures for ensuring the reliability of the comparison results. 1) The three reference land cover products were selected because they include the majority of land cover types used in CALC-2020, so the comparing results can be more robust and less affected by the issue of classification scheme discrepancy. 2) In this paper, the main goal of **Figure 7** is to demonstrate the capacity of typical Arctic landscape characterization by CALC-2020 map, and therefore two situations can be expected. For the first situation, CALC-2020 is directly comparable to at least one reference land cover dataset, given their similar class compositions (**Figure 7(a), (c), (d)**). For the second situation, we may encounter quite different land cover compositions between CALC-2020 and other products (**Figure 7(b)**), which makes direct comparison difficult. In this case, the Google Earth image plays a more important role in mapping performance evaluation. To minimize the confusion, we added a sentence in the revised manuscript (**Page 15**, **Lines 313-314**), offering more explicit information on the logic of **Figure 7** for inter-product evaluation.

I suggest to remove the statistics figure in Figure 9b, because it cannot provide useful information for the CALC-2020 map.

**Response**: Thank you for raising this issue for clarification. We agree that each figure/table in a paper should be informative. The main reason of displaying bioclimate zone level land cover compositions is due to its linkage to climate conditions, which cannot be revealed from the country-level statistics (**Figure 9(a)**). More specifically, **Figure 9(b)** confirmed the strength of temperature in determining circumpolar Arctic land cover composition. As SWI gradually increases (bioclimate zone A~E), we can clearly observe area loss in snow/ice and bare land classes, which replaced by vegetation covers. This result matches well with two previously coarse resolution circumpolar Arctic map (Walker et al., 2005; Raynolds et al., 2019). Based on the analysis shown above and careful consideration, **Figure 9(b)** was retained in the revised manuscript. We modified sentences to avoid confusion (**Page 18**, **Lines 351-352**).

*Reference 1: Walker, D. A., Raynolds, M. K., Daniëls, F. J. A., Einarsson, E., Elvebakk, A., Gould, W. A., Katenin, A. E., Kholod, S. S., Markon, C. J., Melnikov, E. S., Moskalenko, N. G., Talbot, S. S., Yurtsev, B. A. (†), and Team, T. other members of the C.: The Circumpolar Arctic vegetation map, J. Veg. Sci., 16, 267–282, 2005.*

*Reference 2: Raynolds, M. K., Walker, D. A., Balser, A., Bay, C., Campbell, M., Cherosov, M. M., Daniëls, F. J. A., Eidesen, P. B., Ermokhina, K. A., Frost, G. V., Jedrzejek, B., Jorgenson, M. T., Kennedy, B. E., Kholod, S. S., Lavrinenko, I. A., Lavrinenko, O. V., Magnússon, B., Matveyeva, N. V., Metúsalemsson, S., Nilsen, L., Olthof, I., Pospelov, I. N., Pospelova, E. B., Pouliot, D., Razzhivin, V., Schaepman-Strub, G., Šibík, J., Telyatnikov, M. Yu., and Troeva, E.: A raster version of the Circumpolar Arctic Vegetation Map (CAVM), Remote Sens. Environ., 232, 111297, 2019.*

Figure 10 about the feature importance is interesting and vital for developing the CALC-2020, I think the author should explain in more details why the topography had highest importance while the optical features were unimportant.

**Response**: We agree with the suggestion. In the revised manuscript, 1) we clearly mentioned the logic why topography is the most important domain for circumpolar Arctic land cover mapping; 2) we also added descriptions on explaining why spectral features played a secondary role (compared to terrain and SAR features) in this study. Please see **Page 19**, **Lines 376-385** in the revised manuscript.

I suggest that authors shared the validation data in the Data availability

**Response**: We do our best to follow Open Science best practice and aim to make all code and data conditionally available upon publication. The CALC-2020 product, as we mentioned in our paper, is completely open for use via Science Data Bank. All other data used in the study (including training and validation samples) are available from the corresponding authors upon reasonable request. We added descriptions on this issue in the revised manuscript (**Page 22**, **Lines 447-448**).

Line 334, Figure 30 should be 10.

**Response**: Corrected in the revised manuscript.

---

## Author Response (AR2)

**Manuscript essd-2022-224, Second Revision**
**Submitted to: Earth System Science Data**

**CALC-2020: a new baseline land cover map at 10 m resolution for the circumpolar Arctic**

**Chong Liu, Xiaoqing Xu, Xuejie Feng, Xiao Cheng, Caixia Liu, and Huabing Huang**

**Responses to Referee #2**

**December 15 2022**

**General comments**

The manuscript has been improved according to the comments. However, I still have three concerns about the revised manuscript:

**Response**: We would like to thank the reviewer for the encouragement, and pointing out issues that help improve the research and the quality of this paper. According to these comments, we further revised the manuscript and added figures/tables to ensure it meets the quality and high standard of ESSD. Below we provided our detailed responses to the points raised.

**Major comments**

1. The novelty of the method and Dataset is a little weak;

**Response**: We agree that the core of any research lies in its novelty and contributions to the scientific community. With the help from two reviewers and the topical editor, we respectively argue that our revised manuscript has novelty from the following two aspects.

**1)** We proposed a new paradigm for fine resolution Arctic land cover mapping using remotely sensed big data streams. As stated in our paper, the terrestrial Arctic environment is a fundamentally different ecosystem from those at lower latitudes, which calls for reconsideration of every aspect of satellite remote sensing, including data acquisition, land cover legend design, classification, and accuracy assessment. To achieve this goal, we developed a framework that incorporates data from multiple satellites observing the Arctic, and migrated sample from pre-existing knowledge. We modified the FROM-GLC class scheme to represent the biome diversity across the terrestrial Arctic, and used locally adaptive (country-specific) RFC models to reduce the unfavorable impacts incurred by geographical variability. To confirm the reliability of the proposed framework, the classification result was evaluated against validation sample data, in-situ records, and contemporary land cover products.

**2)** More importantly, the developed CALC-2020 dataset can contribute to the scientific community by providing the first spatially continuous map of circumpolar Arctic land cover at 10 m resolution. Based on this map, new insights can be given into Pan-Arctic land cover composition and distribution that have not been fully documented in existing publications or datasets, and therefore advances our understanding of a continuously changing Arctic as well as its global environmental impacts.

Based on the above two aspects, we kindly hope the reviewer re-evaluate our work after its 2rd-round revision.

the validation accuracy of the CALC is still low (shrub tundra, which is the advantage of the dataset).

**Response**: We gave critical thinking and double-check the sample-based validation procedure of CALC-2020. Some key points are clarified as follows.

**1)** Following reviewer#1's suggestion, we adopted "good practices" by Olofsson et al. (2014), and built the error matrix as well as associated sample-based accuracy statistics of the CALC-2020 map. The OA of the CALC-2020 land cover product for the circumpolar Arctic is 79.3±1.0% (95% confidence interval), which is reasonable and comparable to most regional-scale Arctic land cover mapping efforts (typically ranging from 65% to 85%).

**2)** As pointed out by the reviewer, the stratified error-adjusted PA estimate of shrub tundra is 47.1±5.7%, due primarily to the influence of estimation weights (area proportions of map classes). Here we refer to a recently published, continental-scale land cover product using the same "sample-adjusted" validation approach (GLanCE for North America by Friedl et al., 2022), and found a similar PA for the shrub biome (40.8 ± 6.0). It should be noted that all accuracy metrics reported in the manuscript are based on error matrix of area proportion (Olofsson et al., 2014), rather than traditional confusion matrix of sample counts. As shown in **Table R1**, the traditional confusion matrix-derived PA of shrub tundra (SRT) is 68.1%, which is much higher than its stratified error-adjusted estimation. To avoid possible misunderstanding, we added **Table R1** as **Table S3** in **Supplement**, and clarified this point in the revised manuscript:

**Table R1** Accuracy statistics of the CALC-2020 map based on traditional confusion matrix of sample counts. Abbreviations of land cover are given in the main text.

| Class | CRO | FST | GRT | **SRT** | WET | OWT | LAM | MMI | BAR | IAS |
|---|---|---|---|---|---|---|---|---|---|---|
| UA (%) | 93.5 | 75.0 | 81.9 | **84.7** | 81.0 | 93.7 | 76.8 | 95.9 | 75.1 | 74.0 |
| PA (%) | 100.0 | 100.0 | 94.3 | **68.1** | 79.6 | 63.7 | 78.1 | 100.0 | 63.0 | 90.1 |
| OA (%) | | | | | 79.6 | | | | | |
| Kappa | | | | | 0.735 | | | | | |

**Page 11**, **Lines 261-264 in the revised manuscript**: It should be noted that all metrics reported in **Table 3** are based on error matrix of area proportion, therefore inevitably different with those derived from traditional confusion matrix of sample counts. For example, the traditional confusion matrix-derived PA of shrub tundra is 68.1% (**Table S3**), whereas its stratified error-adjusted PA estimate is lower, due primarily to the influence of estimation weights (area proportions of map classes).

**3)** We did recognize the challenge of shrub tundra mapping by the current version of CALC-2020, although it was developed by incorporating Optical, SAR and DEM images. Some researches (e.g., Gong et al, 2016) suggested improved classification performance of shrubland by using the vegetation height as an auxiliary feature metric. However, such a geospatial dataset is currently unavailable for the entire terrestrial Arctic. As a result, we expect a higher, more consistent accuracy level of CALC as new multi-source Earth observation data become available in the future. We clearly highlighted this issue in the **Discussion** section of the revised manuscript:

**Page 20**, **Lines 417-420 in the revised manuscript**: With updates of more multi-source Earth observation data in the future (e.g., vegetation height), the development of Version 2 CALC product will become a possible topic, which is expected to have a hierarchical classification scheme and improved mapping performances for some specific biomes (e.g., shrub tundra).

*Reference 1: Olofsson, P., Foody, G. M., Herold, M., et al.: Good practices for estimating area and assessing accuracy of land change, Remote Sens. Environ., 148, 42–57, 2014.*

*Reference 2: Friedl, M. A., Woodcock, C. E., Olofsson, P., et al.: Medium Spatial Resolution Mapping of Global Land Cover and Land Cover Change Across Multiple Decades From Landsat, Front. Remote Sens., 3, 2022.*

*Reference 3: Gong, P., Yu, L., Li, C., et al.: A new research paradigm for global land cover mapping, Annals of GIS, 22, 87–102, 2016.*

2. The confusion matrix cannot convince me espcially in the confusion between open water, ice and snow and barren land. In my opinion, the open water is easier to confused to the wetland.

**Response**: We have carefully double-checked our sample validation procedure to ensure each element in the error matrix (**Table 3**) correct. Given the generally large reflectance discrepancy between water and non-water covers, the less desirable performance of CALC-2020 in water extraction may seem unexpected. This highlights the distinctiveness of Arctic's geographical environment that can affect the spectral signal of water in space and time (Gong et al., 2016). At high latitudes, shallow water bodies are easily confused with barren lands because of the mixed pixel issue (**FigureR1a~b**). Moreover, the employed satellite images may only capture the freezing stage for some water pixels, which were misclassified as ice/snow in the CALC-2020 map (**Figure R1c**). We added **Figure R1** as **Figure S4** in **Supplement**, and explicitly discussed this issue in the revised manuscript.

[Figure]

**Figure R1** Sentinel-2 true-color images showing three typical examples of water sample points misclassified by CALC-2020. (a) Mixed pixel of lake water and barren land (centred at 69.1°N, 134.8°W). (b) Mixed pixel of river water and barren land (centred at 75.8°N, 99.7°E). (c) Frozen lake water confused with ice/snow (centred at 78.9°N, 20.1°E). The Google Earth VHR image is also displayed for each example.

**Page 11**, **Lines 264-269 in the revised manuscript**: Given the generally large reflectance discrepancy between water and non-water covers, the less desirable performance of CALC-2020 in water extraction may seem unexpected. This highlights the distinctiveness of Arctic's geographical environment that can affect the spectral signal of water in space and time (Gong et al., 2016). Specifically, shallow water bodies are easily confused with barren lands because of the mixed pixel issue (**Figure S4a~b**). Moreover, the employed satellite images may only capture the freezing stage for some water pixels, which were misclassified as ice/snow in the CALC-2020 map (**Figure S4c**).

*Reference: Gong, P., Yu, L., Li, C., et al.: A new research paradigm for global land cover mapping, Annals of GIS, 22, 87–102, 2016.*

3. The comparisons with other global products are unfair, I suggest the authors can provide the quantative metrics, and add some national land-cover products as the comparative datasets to demonstrate the advantages of the CALC-2020.

**Response**: We agree with the reviewer's suggestion. In the revised manuscript, we additionally compared our map with two national-scale land cover products, and calculated accuracy metrics of three global land cover datasets. The summary of major changes are as follows:

**1)** We compared the CALC-2020 map with NLCD (Jin et al., 2017) and Land Cover of Canada (Wulder et al., 2018) because they are currently the only two fine resolution national-scale land cover products covering parts of our study area. Both NLCD and Land Cover of Canada utilize hierarchical classification schemes, we therefore merged their level-2 class legends to that of CALC-2020 (level-1) for interpretation. It should be noted that our stratified validation sample was designed for the circumpolar Arctic, thus not suitable for quantifying the accuracy of national-scale products (Olofsson et al., 2014). As an alternative, we compared the land cover distribution of our estimation with those from NLCD and Land Cover of Canada at 30 m resolution. As shown in **Figure R2**, CALC-2020 map shows overall high agreements with both national-scale land cover products, reflecting its reasonable mapping performance in North America. We added **Figure R2** as **Figure S6** in **Supplement**, and explicitly discussed this issue in the revised manuscript.

[Figure]

**Figure R2** Comparison of CALC-2020 and two national-scale land cover products: NLCD 2016 (a) and Land Cover of Canada 2015 (b). The level-2 classification schemes of two national-scale land cover products are merged to that of CALC-2020 (level-1) for interpretation.

**Page 15**, **Lines 333-336 in the revised manuscript**: The reasonable performance of the CALC-2020 map for North America was also confirmed by referring to two national-scale land cover products: NLCD and Land Cover of Canada, both of which exhibit high agreement of land cover distribution pattern with CALC-2020 at their level-1 classification schemes (**Figure S6**).

**2)** We calculated OA, PA and UA of three global land cover products (ESA WorldCover, ESRI Global Land Cover, GlobeLand30) using the same sample-based evaluation approach applied to CALC-2020. To harmonize various classification legends to that of CALC-2020, the grass (ESA WorldCover, ESRI Global Land Cover) and wet tundra classes (GlobeLand30) were treated as equivalents of graminoid tundra and wetland, respectively. We reported limited classification accuracies of the three compared land cover products (**Figure R3**), with OAs ranging from 48.5% to 71.2%. In the meantime, three global-scale datasets exhibit wide PA and UA variations. The quantitative comparison result is generally consistent with a previous research (Liang et al., 2019), implying imbalanced performances across different Arctic land cover types mapped by pre-existing global-scale datasets. We added **Figure R3** as **Figure 7** in the revised manuscript, and offered relevant descriptions.

[Figure]

**Figure R3** Accuracy statistics of three global land cover products for the circumpolar Arctic based on validation sample. Note that the grass (ESA WorldCover, ESRI Global Land Cover) and wet tundra classes (GlobeLand30) are treated as equivalents of graminoid tundra and wetland, respectively. The dark grey color represents the absence of specific class(es).

**Page 10**, **Lines 237-240 in the revised manuscript**: As an additional comparison to complement the inter-product evaluation, we used the validation sample shown in **Figure S3** to calculate accuracy metrics of three global land cover products. To harmonize various classification legends to that of CALC-2020, the grass (ESA WorldCover, ESRI Global Land Cover) and wet tundra classes (GlobeLand30) were treated as equivalents of graminoid tundra and wetland, respectively.

**Page 14**, **Lines 313-316 in the revised manuscript**: Using the same sample-based evaluation approach applied to CALC-2020, we reported limited classification accuracies of three global

land cover products for the circumpolar Arctic (**Figure 7**), with OAs ranging from 48.5% to 71.2%. In the meantime, these global-scale datasets exhibit wide PA and UA variations, implying imbalanced mapping performances across different Arctic land cover types (Liang et al., 2019).

*Reference 1: Jin, S., Yang, L., Zhu, Z., and Homer, C.: A land cover change detection and classification protocol for updating Alaska NLCD 2001 to 2011, Remote Sens. Environ., 195, 44–55, 2017.*

*Reference 2: Wulder, M. A., Coops, N. C., Roy, D. P., et al.: Land cover 2.0, Int. J. Remote Sens., 39, 4254–4284, 2018.*

*Reference 3: Olofsson, P., Foody, G. M., Herold, M., et al.: Good practices for estimating area and assessing accuracy of land change, Remote Sens. Environ., 148, 42–57, 2014.*

*Reference 4: Liang, L., Liu, Q., Liu, G., et al.: Accuracy Evaluation and Consistency Analysis of Four Global Land Cover Products in the Arctic Region, Remote Sens., 11, 2019.*

---

## Author Response (AR3)

**Responses to Editor:**

For now, we will proceed with your manuscript as submitted. However, please adjust your manuscript files before your next file upload (next round of revision or after acceptance) considering the following requirements

**Response:** Thank you! We address the comments below.

1. A table is included as figure (Figure 7). Please re-label this as table and the references in the manuscript text must be adjusted accordingly. If the colour spectrum of these tables is necessary and cannot be exchanged for footnotes, bold, or italic, then the table must be inserted as an image, but still be called a table.

**Response:** Thanks for the advice. We respectively argue that **Figure 7** cannot be re-labelled as a Table due to the existence of bar chart (left panel). As an alternative, we modified this figure by removing numbers in heatmaps. Please let me know if any further correction is needed.

2. Please ensure that the colour schemes used in your maps and charts allow readers with colour vision deficiencies to correctly interpret your findings. Please check your figures using the Coblis – Color Blindness Simulator (https://www.color-blindness.com/coblis-color-blindness-simulator/) and revise the colour schemes accordingly.

**Response:** Agreed and checked. We revised the color scheme of **Figure 4**, and refilled NA values of **Figure 7** as stripped blocks.

3. Coloured or marked text in *.pdf manuscript file is not allowed. Please provide a clean version of *pdf manuscript file (with black text) with the next revision.

**Response:** Corrected.